# Struct-Bench: A Benchmark for Differentially Private Structured Text Generation

Shuaiqi Wang[1][*][†], Vikas Raunak[2][*][‡], Arturs Backurs[3], Victor Reis[3], Pei Zhou[2][§], Sihao Chen[2], Longqi Yang[2], Zinan Lin[3][§], Sergey Yekhanin[3], and Giulia Fanti[1]

[1]Carnegie Mellon University, Pittsburgh, PA, United States    [2]Microsoft Corporation, Redmond, WA, United States    [3]Microsoft Research, Redmond, WA, United States
[1]{shuaiqiw, gfanti}@andrew.cmu.edu    [2]{viraunak, pei.zhou, sihaochen, longqi.yang}@microsoft.com    [3]{arturs.backurs, victorol, zinanlin, yekhanin}@microsoft.com

## Abstract

Differentially private (DP) synthetic data generation is a promising technique for utilizing private datasets that otherwise cannot be exposed for model training or other analytics. While much research literature has focused on generating private unstructured text and image data, in enterprise settings, *structured data* (e.g., tabular) is more common, often including natural language fields or components. Existing synthetic data evaluation techniques (e.g., FID) struggle to capture the structural properties and correlations of such datasets. In this work, we propose Struct-Bench, a framework and benchmark for *evaluating* synthetic datasets derived from structured datasets that contain natural language data. The Struct-Bench framework requires users to provide a representation of their dataset structure as a Context-Free Grammar (CFG). Our benchmark comprises 5 real-world and 2 synthetically generated datasets. We show that these datasets demonstrably present a great challenge even for state-of-the-art DP synthetic data generation methods. Struct-Bench provides reference implementations of different metrics and a leaderboard, offering a standardized platform to benchmark and investigate privacy-preserving synthetic data methods. We also present a case study showing how Struct-Bench improves the synthetic data quality of Private Evolution (PE) on structured data. The benchmark and the leaderboard have been publicly made available at https://struct-bench.github.io.

## 1   Introduction

Enterprise settings often feature datasets that include both *structured* relationships between fields or objects, and fields that contain *natural language data*. For example, consider a dataset of user queries to a search engine, and the corresponding results. The dataset is structured in a question-and-response format, and both the query and the response contain natural language. Another example consists of patients' medical records, which can include multiple events over time, visits with different providers for different ailments, and associated (natural language) doctors' notes. Such datasets are valuable for downstream use cases (e.g., training predictive models, understanding preferences), but they cannot always be used directly, due to privacy or data use restrictions.

---

[*]These authors contributed equally to this work.
[†]This work was partially done while an intern at Microsoft.
[‡]Now at Google DeepMind.
[§]Primary internship mentors: Pei Zhou, Zinan Lin.

39th Conference on Neural Information Processing Systems (NeurIPS 2025) Track on Datasets and Benchmarks.

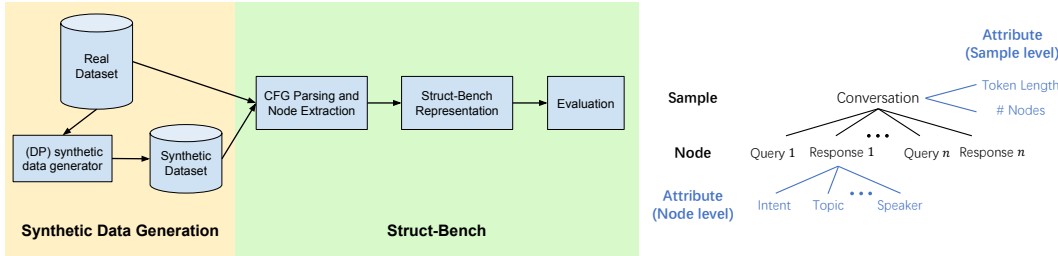

(a) Struct-Bench evaluation pipeline. The synthetic dataset can be generated via DP generation methods with private access to the private dataset. Struct-Bench evaluates a synthetic dataset by parsing samples and extracting nodes and their attributes using context-free grammar (CFG) with full access to both private and synthetic datasets.

(b) Sample level view of the Struct-Bench. As an example, a multi-round conversation is parsed by CFG into several nodes with types Query and Response. Struct-Bench extracts its sample-level and node-level attributes for evaluation.

Figure 1: Dataset level and sample level views into the Struct-Bench framework.

Differentially private (DP) synthetic data generation is an increasingly important technique for making use of such sensitive datasets in machine learning (ML) pipelines [66, 42, 60]. There exist many data synthesis techniques tailored to unstructured data like images or text [54, 27, 60, 16] and tabular data [36, 55, 66]. However, existing synthetic data evaluation frameworks do not naturally capture the salient properties of datasets that feature both general structural properties *and* natural language elements. Evaluation metrics for unstructured data, like Fréchet Inception Distance (FID) [20], or precision and recall [25, 43], do not capture the structural properties of data. In fact, we show that synthetic data that completely fails to capture structural constraints can still achieve high precision and recall scores. For example, the ShareGPT dataset [1], which consists of multi-round conversations between a human and an AI agent, requires the format tokens 'HUMAN:' before queries and 'GPT:' before responses. Consider the synthetic sample: "How are you? I'm doing well." Although semantically reasonable—and therefore scoring high on precision—it violates the required format. This underscores the importance of structure-aware evaluation. On the other hand, evaluation frameworks for tabular data are designed primarily for datasets with categorical or numeric fields [19, 62, 8], and do not naturally extend to natural language fields. For instance, these frameworks often compare $k$-way marginal distributions in the real and synthetic data, which is not tractable for natural language.

In this work, drawing inspiration from Natural Language Generation (NLG) benchmarks such as GEM [12], we present *Struct-Bench*: a composite benchmark and an automatic evaluation protocol that measures the quality of structured, natural language-based synthetic data relative to their corresponding real (possibly private) datasets. In the Struct-Bench framework, a user first selects one or more real datasets for which they want to evaluate a corresponding synthetic dataset. For each dataset, the user provides a set of production rules under the generative grammar formalism (i.e., a context-free grammar [21]); the user then selects *key nodes*, which will be programmatically extracted from the parse tree of dataset samples; key nodes are used to measure important correlations and properties of the synthetic data. Based on this dataset representation, Struct-Bench measures an array of syntactic (i.e., structural) and semantic properties of the synthetic dataset. We illustrate the dataset level and sample level views into Struct-Bench in Fig. 1.

Our contributions are as follows:

1. **Benchmark:** We propose Struct-Bench, a novel evaluation framework and a benchmark to evaluate synthetic data quality relative to a real dataset, where the real dataset features complex inter-field structural relations, and at least some fields contain natural language. A key observation is that many structural properties (and even semantic properties) can be modeled and evaluated with the help of a context-free grammar. We also provide a public leaderboard and a reference, extensible implementation of the Struct-Bench framework, which can be used to benchmark new DP synthetic data generation methods on other datasets in a standardized manner.

2. **Findings:** We use Struct-Bench to benchmark and analyze state-of-the-art (SOTA) DP synthetic data generation techniques on seven diverse datasets, including real-world textual and tabular

datasets, as well as synthetic datasets with controllable data attributes. Our main findings are (1) no single metric fully describes synthetic data quality; (2) none of the existing SOTA DP synthetic data generation techniques are able to reliably capture the structural properties of data without sacrificing semantic performance. Finding (1) highlights the importance of using multiple metrics to evaluate synthetic data, which is a key contribution of our work, and (2) underscores the need for further research in synthetic structured data generation. We also conduct a case study to show how to algorithmically improve on SOTA DP synthetic data generation methods, Private Evolution (PE) [27, 60, 28], using the insights from Struct-Bench. These improvements achieve nearly 100% compliance with dataset structural constraints, while also improving semantic and statistic metrics.

## 2 Struct-Bench Framework and Evaluation Protocol

We aim to design an evaluation framework that measures how closely a real, private dataset $\mathcal{D}$ matches a synthetic dataset $\mathcal{D}'$. The real dataset $\mathcal{D}$ features (1) an inherent *structure*, and (2) *natural language* fields. Our evaluation framework must therefore quantify how well $\mathcal{D}'$ has acquired the structure and content of the private dataset. As these are not well-defined quantities, we define a framework for representing a real dataset $\mathcal{D}$, as well as a suite of metrics to capture how well the synthetic dataset matches the syntax and semantics of the real dataset.

**Notation** Consider a dataset $\mathcal{D} = (D_i)_{i=1}^m$ with $m$ samples. As a concrete example, suppose $\mathcal{D}$ is the ShareGPT dataset [1], which contains multi-round conversations between a human and an AI agent. We illustrate a sample of ShareGPT in §B.1. A dataset is a set of *samples*; in ShareGPT, each sample $D_i \in \mathcal{D}$ is one full conversation (e.g., a few iterations of conversation between the human and the agent). Each sample $D_i$ contains a set of nodes $O_1, O_2, \ldots, O_{n_i}$ (the number of nodes can vary across samples). In the ShareGPT example, each node $O_j$ is one text snippet—either a single query from the human or a response from the AI agent, as illustrated in Fig. 1b.

Samples and nodes can have *attributes*, which are either numeric or categorical. Each sample $D_i$ is associated with *sample-level attributes* $a_1, \ldots, a_m$. These can be any derived property of the whole sample, such as token length of the conversation in ShareGPT. A node $O_j$ can also have *node-level attributes* $v_1, \ldots, v_{n_j}$. In ShareGPT, attributes could include the token length of a node, the identity of the speaker (agent or human), and the topic of the query or response (obtained through human or LLM-based labeling).

**Dataset Representation** The nodes of a dataset may satisfy complex structural relations. For example, in ShareGPT, each node can only have `human` or `agent` as its `speaker` attribute, and successive nodes should always alternate speaker between `human` and `agent`. This information is considered public (to the synthetic data holder); for example, if an enterprise is training a DP synthetic dataset to model a private dataset of search engine queries, the enterprise is likely to know the *schema* of the data, even if they do not know the contents. Such relations are captured in Struct-Bench by a *context-free grammar (CFG)*, which specifies different categories of nodes and the relations between them. As an example, we provide the CFG of ShareGPT in §B.1. To add a new dataset to Struct-Bench, a user must write structural dependencies that should be enforced in the form of a CFG.[1] For each dataset (real or synthetic), Struct-Bench then uses the dataset-holder-provided CFG to construct a parse tree for each sample.

*Remark:* An alternate design choice could be to specify each dataset under the formalism of *context-sensitive grammars (CSGs)*, which are more expressive than CFGs and can capture semantic dependencies. Specifying a CSG, however, requires significantly more domain knowledge and detail, making it a more burdensome—and potentially error-prone—process than specifying a CFG. Hence, instead of encoding semantic dependencies as hard constraints via CSG, we empirically assess them by introducing a metric Key Node Dependency (KND) in §2.1. KND is simple to implement, requires less domain expertise than a full CSG specification, and statistically captures correlations in the data.

Once the CFG is defined, the user also specifies a set of *key nodes*. For instance, if the CFG defines a set of node types $\mathcal{S} = \{\text{Query}, \text{Response}, \text{Follow-Up}\}$, the set of key nodes $\mathcal{K}$ can be any subset of $\mathcal{S}$. Key nodes are expected to exhibit strong dependencies in the data. For example, in ShareGPT, we would define each query-response as a pair of key nodes. We will assess whether the correlations

---

[1]The CFG specification only needs to be performed manually once when the dataset is onboarded, and it may be possible to leverage LLMs to summarize the structure and generate the CFG automatically [53].

among these key nodes are preserved in the synthetic data, as detailed later in §2.1. If a user does not specify key nodes, all nodes are treated as key nodes.

**Privacy Constraint**   In this work, our goal is to evaluate differentially private synthetic data generation algorithms. A data generator $\mathcal{M}$ is $(\epsilon, \delta)$-differentially-private if for any neighboring datasets $\mathcal{D}_0$ and $\mathcal{D}_1$ (i.e., $\mathcal{D}_0$ and $\mathcal{D}_1$ differ one sample), and any set $S \subseteq range\left(\mathcal{M}\right)$, we have

$$\mathbb{P}\left(\mathcal{M}\left(\mathcal{D}_0\right) \in S\right) \leq e^\epsilon \cdot \mathbb{P}\left(\mathcal{M}\left(\mathcal{D}_1\right) \in S\right) + \delta \ .$$

Intuitively, the output synthetic data distribution should not depend too much on any single sample in the input dataset. Today, there exist many algorithms for generating DP synthetic data (some of which can accommodate text and/or structured data) [63, 60, 22, 49, 51]. Struct-Bench provides a systematic way of comparing DP synthetic data generators.

Struct-Bench can also be applied to other forms of privacy-preserving synthetic data, such as those generated under frameworks like quantitative information flow [46], statistical maximal leakage [31, 58], and distribution privacy [48], among others. To ensure a fair comparison, all synthetic data generation baselines should be compared under the same privacy framework.

*Remark:* Private DP data synthesis and non-private data synthesis are two problems with different applications and objectives. In the private setting, the goal is typically to match synthetic data to a private dataset as closely as possible under a DP constraint [51, 42, 29, 22, 60]. On the other hand, if there is no privacy constraint and if one wants to match the real data as closely as possible, one should just use the real data. Indeed, in the non-private setting, synthetic data is typically designed to deviate from the "real" dataset (e.g., conditional generation of a specific class of data) [18, 35, 9, 10, 11, 30]. In our design and evaluation, we focus on DP synthetic data, so Struct-Bench is designed to measure similarity between synthetic and real datasets. However, the metrics in Struct-Bench could be helpful for benchmarking non-private synthetic data as well; we leave this to future work.

**Summary of Inputs**   In summary, the Struct-Bench framework takes as input: (1) a real dataset $\mathcal{D}$,[2] (2) a synthetic dataset $\mathcal{D}'$, (3) a CFG that represents the structural characteristics of the data, (4) a set of key nodes (optional) from the CFG, which represent the types of nodes whose correlations are important. Given these inputs, Struct-Bench automatically calculates the following suite of metrics.

## 2.1 Struct-Bench Metrics

We report three types of metrics: structural, non-structural, and downstream task accuracy.

**Structural Metrics**   These are metrics that depend on the CFG in some way. Within a sample, structure can be defined at the level of the whole sample (*CFG Pass Rate*, *Attribute Match*), groups of nodes (*Key Node Dependency*), and individual nodes (*Attribute Match*). (1) **CFG Pass Rate (CFG-PR)** measures the fraction of samples in the synthetic dataset $\mathcal{D}'$ that parse correctly under the CFG. (2) **Key Node Dependency (KND)** measures the semantic dependencies between "key node pairs", which are pairs of nodes believed to have a meaningful relation;[3] for example, in a question-and-answer dataset, we would expect that each associated question and answer pair should have a strong correlation. Programmatically, users specify pairs of key nodes with `Tregex` [26], a tool for matching regular expressions on trees. Typically, we can measure the dependencies by cosine similarity of the node embeddings, while one can also adopt LLM as a judge or other dependency functions defined by the users. For a dataset, we can construct a distribution of dependencies of node pairs in the same pattern. To evaluate the similarity of the node dependencies captured by the private and synthetic dataset, we then calculate the distributional distance, e.g., Wasserstein-2 distance, between the private and synthetic dependency distributions. (3) **Attribute Match (AM)** measures the distributional distance of sample-level attributes (e.g., number of nodes) or node-level attributes (e.g., node token length) between the private and synthetic datasets. The attribute can be a statistical property or a semantic property, and it can be derived either from an explicit attribute function or through human annotation or LLM-based labeling. The distributional distance can be Wasserstein-2 distance if the attribute is numeric or total variation distance if the attribute is categorical. Precise definitions and instantiation guidelines for KND and AM are in §A. For structural metrics, higher CFG-PR is better, whereas lower KND and AM indicate better performance.

---

[2]The input data should be in string format. For structured data with categorical or numerical values, we convert it to a JSON object where the attribute of each column is the key, and the actual value/text is its value.

[3]We measure dependencies between node pairs and do not consider higher-order relationships, as they are more computationally intensive and the dependency functions can vary across different scenarios.

Table 1: List of datasets with descriptions, key nodes, and sizes of real and generated data.

| Dataset | Description | Key Nodes | Number of Samples | |
|---|---|---|---|---|
| | | | Real | Generated |
| ShareGPT | Human–GPT conversations | query, response | 3 000 | 600 |
| ICLR | ICLR paper reviews & rebuttals | review, rebuttal, comment | 3 000 | 300 |
| Adult | Census dataset | native country, workclass | 50 000 | 31 561 |
| Water | Water-bottle reviews | title, review | 25 000 | 20 000 |
| Arena | Chatbot-Arena conversations | conversation 1 & 2 | 25 000 | 20 000 |
| Reviews | Synthetic product reviews | review, rating | 2 000 | 2 000 |
| Grounding | Synthetic grounded QA | query, response | 2 500 | 2 000 |

**Non-Structural Metrics**   Following prior precedent [51, 5], non-structural metrics are per-sample metrics that do not rely on the CFG, i.e. they are unrelated to the structure of the data. These metrics quantify the similarity between the content of the generated synthetic data and the real/private data. Moreoever, as in prior works on unstructured DP synthetic data [25, 60], we report the precision (**KNN-Precision**) and recall (**KNN-Recall**) for each synthetic dataset. Roughly, KNN-Precision (resp. KNN-Recall) calculates the proportion of synthetic (resp. private) samples whose embedding distance to a private (resp. synthetic) sample is smaller than this sample's $k$-th nearest neighbor within its own dataset. KNN-Precision evaluates the average semantic quality of the synthetic samples, and KNN-Recall assesses the semantic diversity of the samples [25].

**Downstream Evaluations (DE) Based on Label Prediction**   The eventual goal of synthetic data is typically a **Downstream Task**, e.g., training a machine learning (ML) model. Struct-Bench allows users to design their own downstream label prediction tasks. Our pipeline includes label generation, downstream model training, and evaluation. *(1) Label Generation*: The evaluation pipeline requires labels both for the synthetic and real data. Labels can either be generated by a human or we can use large language models (LLMs) to simulate a human labeler. In our evaluation, we adopted GPT-4o to label the samples, but the Struct-Bench codebase gives users flexibility to choose a different LLM. We provide guidelines for the label generation process in §A.3. *(2) Downstream Model Training*: To conduct label prediction, we fine-tune a language model based on the synthetic dataset and its label. Since the samples may have long text, we adopt Longformer [7] in this paper. *(3) Evaluation:* We evaluate the downstream task by calculating the prediction accuracy (Acc) on a held-out test set from the real data. This is commonly done in synthetic data evaluations, and is known as the train-synthetic-test-real (TSTR) framework [56, 39, 30].[4]

## 3   Benchmarking Differentially Private Synthetic Data

To demonstrate the utility of the Struct-Bench framework, we instantiate and implement it on a set of seven datasets and four DP synthetic data generation methods.

### 3.1   Struct-Bench Datasets

We include three types of datasets in Struct-Bench (details in §B.2).

**Real-World Datasets with Graph-Structured Dependencies**   We use **ShareGPT** [1] and **ICLR** 2024 paper reviews [2], two real-world graph-structured datasets that differ in (1) *Content*: multi-round user–GPT conversations vs. author–reviewer reviews, rebuttals, and comments; (2) *topic*: open-domain vs. AI-research–specific; and (3) *structure*: linear query–response chains vs. tree-structured threads (multiple reviews, rebuttals, and follow-up discussions). Both offer diverse semantics, clear structure, and at least two node types. Notably, the ICLR 2024 dataset was released after the training data cut-off date for the LLMs we evaluate.

**Real-World Tabular Datasets**   Although there exist benchmarks for tabular data [5, 51], they do not naturally extend to natural language fields. We evaluate Struct-Bench on three tabular datasets: **Water** [52] and **Arena** [67], which include textual fields, and **Adult** [6], which contains only numeric and categorical values. We include Adult to demonstrate that Struct-Bench can be applied to non-textual

---

[4]Some evaluation frameworks compare TSTR with train-real-test-real (TRTR) for a self-contained evaluation [64]. However, since we are comparing different synthetic data generation algorithms *against each other*, we compute only TSTR in this case, which is slightly more interpretable.

data as well, though this data type is not the focus of our work. Water comprises water-bottle reviews, Arena pairs human–model conversations, and Adult provides census records.

**Synthetic Datasets with Controllable Data Attributes**  To explicitly control structural and semantic complexity, we create two datasets: **Synthetic Reviews**, with reviews varying in sentiment and scores, and **Synthetic Grounding**, with source documents paired with question–answer tasks.

We show the key nodes of each dataset and the sizes of the real and generated data in Table 1, and defer the detailed data modeling to §B.3.

## 3.2  Struct-Bench Synthetic Data Generation Baselines

Since we focus on structured datasets that contain natural language, we select LLM-based DP synthetic data generators as our baselines, specifically, Private Evolution (PE) [27, 60, 28], DP model fine-tuning [63, 59, 65], and some variants. While several methods have been proposed to generate DP tabular data using LLMs [4, 55], they are typically limited to handling only numerical or categorical values and cannot generate structured data that incorporates natural language.

**Private Evolution (PE) [27, 60, 28]**  PE is a leading training-free DP synthetic data generation algorithm that makes use of foundation models pre-trained on public data [27, 60, 28, 22, 16, 57]. PE first uses a Random API to generate initial samples from the foundation language model. Then, it iteratively: (1) constructs a differentially private (noisy) voting histogram based on private samples voting for their nearest synthetic counterparts; (2) draws samples according to this histogram; and (3) creates new samples with a Variation API that generates perturbed versions of the original samples. We use the PE variant, Augmented Private Evolution (Aug-PE) [60], provided by the Private Evolution library;[5] our only modification is to choose the prompt for the Random and Variation APIs.

**Instruction Following (IF)**  IF prompts a foundation model with the target structure and uses no private data ($\epsilon = 0$). It is effectively a zero-shot PE, i.e., equivalent to using only Random API in PE.

**DP Fine-Tuning (DP-FT) [63, 59, 65]**  DP-FT adopts DP stochastic gradient descent (DP-SGD) [3] to fine-tune the language model on the next token prediction task. We generate synthetic data unconditionally from the fine-tuned model. This simple baseline remains competitive when training is allowed [16]. Since PE generates synthetic data based on the instructions in a prompt in the Random and Variation APIs, we also include a variant of DP-FT that conditionally generates samples according to the same instructions after DP fine-tuning, which we refer to as *Instruct DP-FT*. We provide details of this instruction fine-tuning in §C.

**Real Data Fine-Tuning (FT)**  FT fine-tunes models without privacy guarantees ($\epsilon = \infty$), serving as a best-case reference for DP-FT. We also include *Instruct FT* with instruction-conditioned generation.

Since DP-FT and FT require model training, we are limited to open-source models and thus use only GPT-2. In contrast, PE and IF do not use model weights, and require only API access. We evaluate them on both GPT-2 and the state-of-the-art GPT-4o. We further evaluate our baselines on additional foundation models in §C.3. PE is run for 10 iterations, and DP-FT and FT are run for 20 epochs. We take $\epsilon \in \{1, 2, 4\}$ for PE and DP-FT, and $\delta = 0$.

## 3.3  Experimental Results

We present the results of benchmarking the DP synthetic data generation methods under ShareGPT and ICLR datasets with $\epsilon = 4$ in Table 2, and defer the results on other datasets to §C.1. We illustrate the performance of the baselines on CFG-PR and KNN-Recall across all datasets in radar plots shown in Fig. 2. We specify the metrics reported and the whole set of evaluation metrics we adopt in §B.4; additional results for $\epsilon = 2$ and $\epsilon = 1$ the on the ShareGPT and ICLR datasets can be found in §C.2. Table 2 and Fig. 2 highlight several main takeaways:

- *No single metric fully describes synthetic data quality.* For a single algorithm and dataset, some metrics can be high, while others remain low (e.g., see CFG-PR and KNN-Recall in Fig. 2). This further motivates the need for Struct-Bench, which aggregates many diverse metrics.

---

[5] https://github.com/microsoft/DPSDA

- *Existing DP synthetic data generators struggle to learn complicated data structures.* All baselines achieve a CFG-PR score below 0.2 on the ICLR dataset, which features more node types and a significantly more intricate graph structure than ShareGPT.

- *DP fine-tuning alone cannot learn structure.* At $\epsilon = 4$, it achieves a CFG-PR of 0 on all of our datasets. Even at $\epsilon = \infty$, it fails to learn structural information on all datasets except ShareGPT, where it achieves a CFG-PR of 0.53; this is likely because ShareGPT contains fewer formatting tokens compared to other datasets (e.g., JSON tags in tabular datasets).

- *PE and IF learn structure at the expense of semantic performance.* Although PE and IF reliably capture data structure with SOTA models, they have poor semantic performance (low KNN-Recall).

- *The performance gap between PE and DP-FT may arise from the foundation models they employ and the use of instruction-guided generation.* Unlike DP-FT, PE does not require model training, which allows us to leverage SOTA models in the Random and Variation APIs. In contrast, DP-FT relies on fine-tuning, and we thus need to use smaller, open-source models due to computational restrictions and the weight access requirement. Nevertheless, with instruction-guided conditional generation, Instruct DP-FT achieves performance comparable to PE across most metrics on both ShareGPT and ICLR, when both use the same foundation model (GPT-2).

Table 2: DP synthetic data generation benchmarking results on Struct-Bench with $\epsilon = 4$. All baselines use GPT-2 unless otherwise specified.

| Dataset | Baseline | Structural Metrics | | | Non-Structural Metrics | | DE |
| | | CFG-PR ↑ | KND ↓ | AM ↓ | KNN-Precision ↑ | KNN-Recall ↑ | Acc↑ |
| --- | --- | --- | --- | --- | --- | --- | --- |
| ShareGPT | IF ($\epsilon = 0$) | 0.03 | 0.07 | 41.86 | 0.64 | 0.31 | 0.28 |
| | IF ($\epsilon = 0$) (*GPT-4o*) | **0.87** | 0.06 | 43.85 | 0.72 | 0.26 | 0.38 |
| | FT ($\epsilon = \infty$) | 0.53 | **0.03** | 52.70 | 0.76 | **0.66** | 0.37 |
| | Instruct FT ($\epsilon = \infty$) | 0.59 | 0.04 | **30.70** | 0.80 | 0.54 | 0.37 |
| | DP-FT | 0 | - | - | 0.02 | 0 | - |
| | Instruct DP-FT | 0.55 | 0.18 | 32.59 | 0.77 | 0.31 | 0.35 |
| | PE | 0.57 | 0.12 | 34.51 | 0.78 | 0.33 | 0.38 |
| | PE (*GPT-4o*) | 0.86 | 0.07 | 38.17 | **0.81** | 0.15 | **0.39** |
| ICLR | IF ($\epsilon = 0$) | 0.09 | **0.11** | 207.62 | 0.66 | 0.28 | 0.39 |
| | IF ($\epsilon = 0$) (*GPT-4o*) | 0.17 | 0.26 | **204.80** | 0.84 | 0.03 | 0.47 |
| | FT ($\epsilon = \infty$) | 0 | - | - | 0.71 | **0.47** | 0.46 |
| | Instruct FT ($\epsilon = \infty$) | 0.09 | 0.16 | 208.37 | 0.77 | 0.29 | 0.51 |
| | DP-FT | 0 | - | - | 0 | 0 | 0.18 |
| | Instruct DP-FT | 0.08 | 0.22 | 237.52 | 0.49 | 0.18 | 0.40 |
| | PE | 0.10 | 0.20 | 248.73 | 0.49 | 0.20 | 0.41 |
| | PE (*GPT-4o*) | **0.19** | 0.26 | 240.94 | **0.98** | 0.02 | **0.52** |

## 4 Case Studies

In this section, we demonstrate how users can leverage Struct-Bench to better understand and improve PE. We focus on PE because, unlike DP-FT, it does not require *training* on private data, offering both efficiency and qualitative advantages [60, 22]. We use the the ShareGPT dataset for this case study. While GPT-4o might achieve stronger performance, we adopt Llama3-8b as the foundation model in this section since it is more cost-efficient for our experiments and, importantly, more affordable and accessible to end users.

We mainly focus on improving structural validity and semantic diversity of the PE synthetic data in this section, and defer a more thorough analysis as well as methods on improving node dependency (KND) to §D.

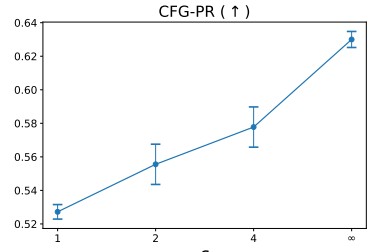

Figure 3: CFG-PR of vanilla PE on ShareGPT. Each data point is averaged over three independent trials. CFG-PR is low for all $\epsilon$.

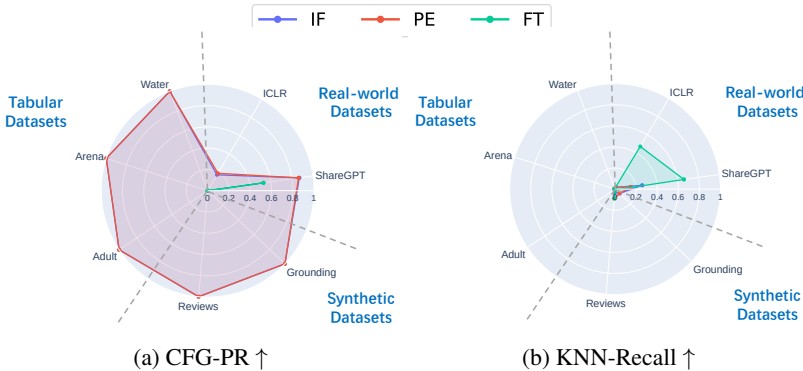

(a) CFG-PR ↑  (b) KNN-Recall ↑

Figure 2: CFG-PR and KNN-Recall of baselines on Struct-Bench with different datasets. While frontier models capture the syntactic structure (high CFG-PR), existing DP synthetic data generators fail on semantic diversity (KNN-Recall). With GPT-4o, IF and PE achieve perfect CFG-PR on tabular and synthetic data, while FT with GPT-2 fails to learn the format. All baselines show near-zero KNN-Recall on tabular and synthetic datasets. We exclude DP-FT because it scores zero on both metrics. Instruction-tuned DP-FT and FT may perform better, as noted in Table 2.

**Problem 1: Structural Validity (CFG-PR) is low.** Structural validity, i.e., CFG-PR, is a critical metric, as many downstream applications on structured datasets expect data to be formatted in a particular way for compatibility with utilities and dataset-specific pipelines. Fig. 3 shows that the CFG-PR of vanilla PE is below 60% when $\epsilon \leq 4$, and only achieves $\sim 63\%$ in the non-private setting (i.e., we run PE with no added noise). This suggests that with smaller foundation models (e.g., Llama3-8B), vanilla PE fails to capture even simple structural constraints.

**Solution 1: LLM-Assisted reformatting can improve CFG compliance.** To improve structural validity, we introduce a *reformatting* feature to the Random and Variation APIs by prompting

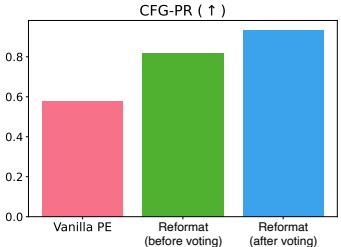

Figure 4: CFG-PR of vanilla PE and PE with reformatting, $\epsilon = 4$.

LLMs to explicitly check and reformat CFG-invalid samples (see §D.2). For example, if PE generates 'HUMAN: How are you?', the model detects the missing response and reformats it to 'HUMAN: How are you? GPT: I'm fine.'

As described in §3.2, PE iteratively generates candidate samples and uses private voting to select the highest-quality ones. Sample reformatting can happen before or after the PE private voting process. We compare both reformatting methods with vanilla PE at $\epsilon = 4$ in Fig. 4. We see that the CFG-PR of both our methods increases by over 20% compared with vanilla PE, and reformatting after the private voting process does the best.

Reformatting can help enforce structural correctness but may distort semantics. For example, as illustrated in Fig. 6, if PE generates "How are you? I'm fine. Thanks.", the model may detect missing format tokens and reformat it to "HUMAN: How are you? I'm fine. GPT: Thanks!" While the reformatted version follows a valid structure, its semantics are flawed—the user's query includes part of the response. Thus, reformatting-before-voting can bias voting against such semantically distorted samples. In contrast, reformatting-after-voting directly reformats only the most highly-voted samples, which are either used as a final output (at the last iteration of PE) or used as seeds in the next iteration.

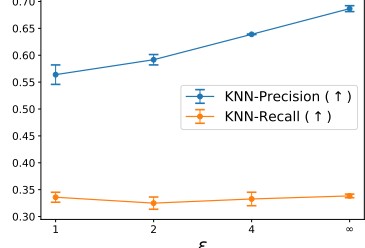

Figure 5: KNN-Precision and KNN-Recall of vanilla PE on ShareGPT under different privacy guarantees. Both are low.

**Problem 2: Semantic diversity (KNN-Recall) is low.** As described in §2.1, KNN-Precision measures the semantic quality of the generated samples, and the KNN-Recall measures how well the semantic diversity of the private dataset is captured in the synthetic data. As we see in Fig. 5, as $\epsilon$ increases, i.e., with looser privacy constraints, the KNN-Precision of vanilla PE increases from 0.56 to 0.69, increases from 0.56 to 0.69,

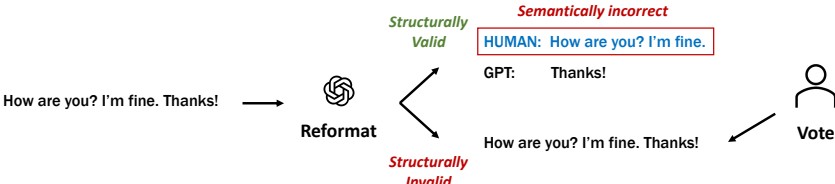

Figure 6: Illustration of reformatting-before-voting on the ShareGPT dataset. The syntactically-correctly reformatted sample follows a valid structure but has flawed semantics—the user's query includes part of the response. The incorrectly reformatted sample preserves semantic integrity; the voting process can select samples that are semantically consistent but structurally invalid.

while KNN-Recall remains very low, around $0.35$. This suggests that vanilla PE focuses on semantic quality while sacrificing diversity.

**Solution 2: Node extraction & auto-generation can improve semantic diversity.** In the Variation API, PE generates new samples by first masking a subset of the original text and then using the LLM to fill in the blanks based on the remaining context. This process largely preserves the original meanings, which limits semantic diversity. For example, if a conversation is about weather, and its masked version retains keywords like 'cloudy' or 'rainy', the blank-filled new sample will likely still be about weather, rather than an unrelated topic like dogs.

To improve semantic diversity, we *extract* specific nodes for blank-filling, and use the LLM to conditionally generate the remaining nodes. We call this "Node extraction and auto-generation" (example on the ShareGPT dataset in Fig. 7). The user specifies which nodes to extract—for instance, roots in the parse tree of the CFG. The remaining nodes are generated by asking an LLM to generate the remaining nodes in the sample, given only the extracted nodes (with blank-filled variations). This pipeline incurs no additional privacy cost due to the post-processing property of DP.

We compare the performance of vanilla PE and PE with node extraction in Fig. 8. Recall that the ShareGPT dataset has only two types of nodes: query and response nodes. In Fig. 8, we consider two variants of the node extraction method: (1) extract all queries and auto-generate all responses (listed as Extract Query), and (2) extract all responses and auto-generate all queries (shown as Extract Response). We show three metrics: KNN-Recall, KNN-Precision, and CFG-PR.

Fig. 8 shows that Extract Query not only improves the KNN-Recall but also the KNN-Precision, (i.e., semantic quality) as auto-generation also ensures the semantic meaning is more consistent and natural across nodes. However, Extract Response does not improve KNN-Recall as the semantic diversity depends mainly on responses, while queries are fairly constrained for a given response. This indicates that the type of nodes extracted is crucial to the performance of semantic diversity.

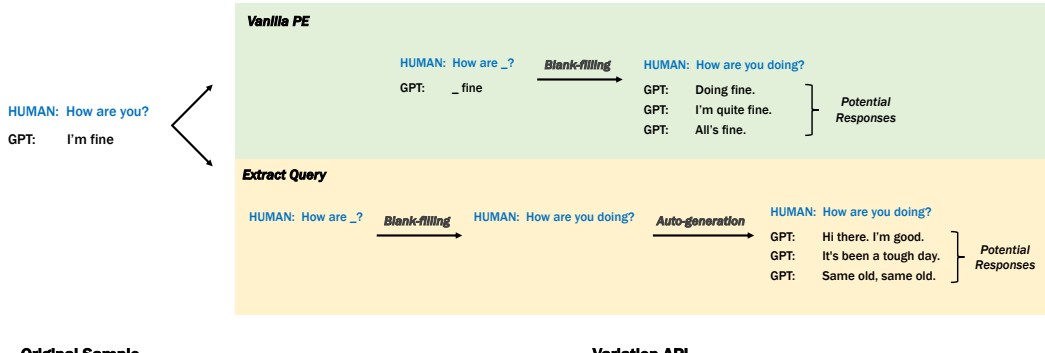

Figure 7: Example executions of the Variation API of vanilla PE and PE with node extraction and auto-generation on ShareGPT. Vanilla PE masks parts of the original text and has the LLM fill the blanks from the remaining context. In our variant, we first extract a node (here, the "Query"), perform blank-filling only on that node, and then generate responses conditioned on the query. By imposing fewer semantic constraints, this approach yields more semantically diverse samples.

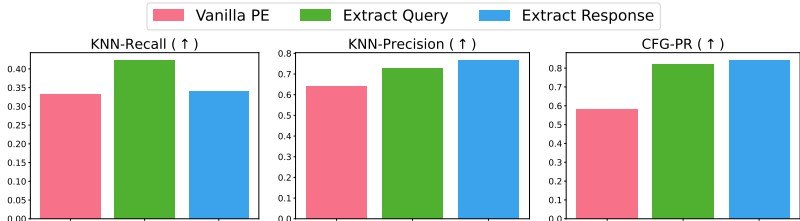

Figure 8: KNN-Precision & KNN-Recall and CFG-PR of vanilla PE and PE with node extraction.

Additionally, since the formatting tokens around the extracted critical nodes will not be accidentally modified by Variation API, the CFG-PR of PE with node extraction is also higher than vanilla PE.

**Combination of our solutions achieves the best performance on most metrics.** We finally compare the performance of our proposed methods. In Fig. 9, we plot the performance of vanilla PE, PE + Reformat, PE + Extract Query, and PE + Reformat + Extract Query. To better visualize differences in the performance of different methods,

we scale the metrics in these radar plots as follows: We assign a score of 0 if CFG-PR=0 or a structure-related metric is not applicable for the dataset, and rescale the values of other metrics from 20 to 100, where 20 indicates the worst performance among all methods, and 100 indicates the performance upper bound the synthetic data can achieve (e.g., CFG-PR=1 or AM=0).

The combination of our proposed methods (orange curve) significantly improves in CFG-PR, achieving up to 94%, and it also outperforms or performs comparably to other methods in most metrics, including semantic metrics (KNN-Precision, KNN-Recall), and statistic metrics (AM on response length and node count).

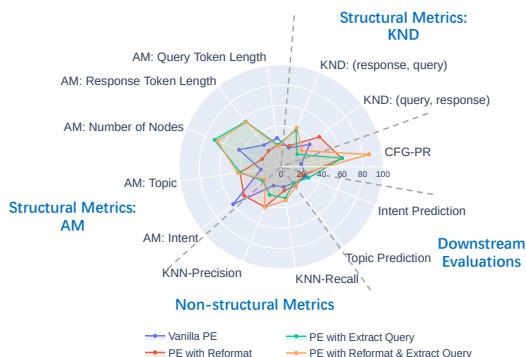

Figure 9: Performance of Different Methods on ShareGPT with $\epsilon = 4$.

## 5   Related Work

**Differentially Private (DP) Synthetic Data Generation**   DP synthetic data generation has developed as an effective tool in the machine learning model development pipeline [66, 42, 27, 60, 28, 55], especially with the advent of instruction-following LLMs generating natural and fluent text [4, 60, 38, 17, 50]. Prior methods, such as Private Evolution (PE) [27, 60, 28, 22, 68, 23, 57], leverage pretrained models (e.g., large language models) or non-neural approaches (e.g., computer graphics tools) for DP synthetic data generation. [27, 60, 28] show that PE could be competitive with DP fine-tuning baselines [63, 59, 65, 13] while does not require training on the private data.

**DP Synthetic Data Evaluation**   Evaluating DP synthetic data presents a unique challenge, namely of quantifying adherence of the synthetic data to arbitrary private datasets [51, 44]. Several benchmarks have been proposed to evaluate DP synthetic data in image [16, 24, 32, 61], text [40, 45, 64], tabular [42, 14, 5, 37, 34, 33], time series [47], and graph data [15]. However, these benchmarks either do not explicitly consider data structure, or their evaluation is confined to numerical or categorical data types, thus limiting their scope.

## 6   Conclusion

In this work, we proposed a new benchmark for DP synthetic data generation named Struct-Bench. To the best of our knowledge, Struct-Bench is the first benchmark to comprehensively evaluate DP synthetic data derived from structured datasets that contain natural language data. Struct-Bench also has the strength of being a composite benchmark, wherein a diverse collection of datasets might preclude algorithmic research to overfit to only a few data types. Through our evaluations, we also characterize the limitations of existing SOTA DP synthetic data generation methods and conduct a case study to show how to improve on SOTA methods using the insights from Struct-Bench.

## Acknowledgements

The authors would like to thank Sivakanth Gopi for his helpful suggestions. This work was supported in part by the National Science Foundation under grants CCF-2338772 and CNS-2148359.

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

# A  Metric Definitions and Instantiation Guidelines

## A.1  Key Node Dependency (KND)

**Definition**  KND measures the distributional distance of node pair dependencies between the synthetic and original data. For a key node pair $(O_i, O_j)$, let $C_{i,j}$ be the cosine similarity between their embeddings, and let $\omega_{C_{i,j}}$ and $\omega'_{C_{i,j}}$ be the distributions of these similarities in the original and synthetic data, respectively. Then, KND is defined as:

$$\text{KND}(O_i, O_j) = Dis(\omega_{C_{i,j}}, \omega'_{C_{i,j}}),$$

where $Dis$ is the Wasserstein-2 distance.

**Instantiation Guideline**  We allow the user to specify key nodes. If not specified, all nodes parsed by CFG are treated as key nodes by default. To instantiate key nodes, we recommend users ask the question *"Which nodes are central to our downstream tasks, and which nodes are semantically related to them?"*. For example, key node pairs could be a query and response in a conversation dataset, or a review and its rating in a product review dataset. We've specified the key nodes of our datasets in Table 1.

## A.2  Attribute Match (AM)

**Definition**  AM calculates the distributional distance of a given attribute between the synthetic and original data. For attribute $a$, let $\omega_a$ and $\omega'_a$ denote its distributions in the original and synthetic data, respectively. Then, AM is defined as:

$$\text{AM}(a) = Dis(\omega_a, \omega'_a).$$

For distributional distance $Dis$, we use Wasserstein-2 distance for numeric attributes and total variation distance for categorical attributes.

**Instantiation Guideline**  Users can specify semantic or statistical attributes. A guiding question is: *"Which data properties matter for our downstream tasks?"* Common semantic attributes include topic, intent, and sentiment; statistical attributes include token length (overall or per node). Original categorical/numerical values are also often relevant. The selected attributes for our datasets are detailed in §B.4.

## A.3  Downstream Evaluations (DE)

**Label Generation Guideline**  To instantiate labels for downstream evaluation, we recommend that users first determine whether the label should be extracted directly from the sample or generated. Extracted labels typically include original categorical or numerical values—such as income level in our Adult dataset or paper decision in the ICLR dataset. If the label is to be generated, we suggest considering the question: *"What is a data property that is present and can be inferred from each node in a sample and is relevant to our downstream tasks?"* Generated labels are usually semantic attributes of the samples or nodes, such as topic, intent, or sentiment. To discourage prompt gaming by data uploaders, we provide a suggested prompt template for labeling: *"For each sample in ____ scenarios, label the ____ property. Possible answers include {____, ____, ____, ...}."* The specific labels used for our datasets are detailed in §B.4.

# B  Data Modeling and Evaluation Items of Each Dataset

## B.1  Examples on ShareGPT

To support the running example of data modeling and representation discussed in §2, we first show a sample of the ShareGPT dataset, the data modeling of it, and the context-free grammar (CFG) according to its data structure.

**A sample of ShareGPT**

```
1  HUMAN: Pretend you are a successful and very thorough real estate
       investor and financial analyst
2  GPT: Sure, I can help you with that. As a successful real estate
       investor and financial analyst, I have a deep understanding of the
        real estate market and the financial factors that drive its
       performance. Here are some key things that I would keep in mind
       when analyzing investment opportunities:...
3  HUMAN: Assume the debt down payment is 50%
4  GPT: Sure, I can adjust the calculations to assume a 50% down payment.
        Here is how the projections would change:...
5  HUMAN: What is the IRR from the perspective of the equity
6  GPT: To calculate the IRR from the perspective of the equity, we need
       to adjust the cash flows to reflect the equity portion of the
       profits. Here is how we can calculate the equity IRR:...
```

**Illustration of the data modeling of ShareGPT**

We illustrate the data modeling of ShareGPT in Fig. 10.

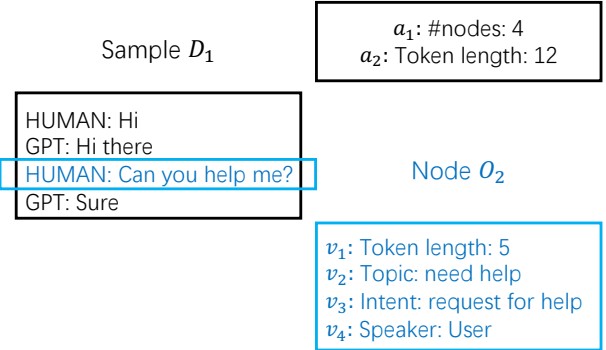

Figure 10: Illustration of the data modeling of ShareGPT.

**CFG of ShareGPT**

```
1  ShareGPT: conversation (conversation)*
2      // ShareGPT contains one or more conversation rounds
3  conversation: query response
4      // Each conversation round contains a query and a response
5  query: "HUMAN:␣" query_text
6      // The query starts with "HUMAN:␣"
7  response: "GPT:␣" response_text
8      // The response starts with "GPT:␣"
9  query_text: /(?s).+?(?=(?:GPT: |$))/
10      // The query text ends before "GPT:␣" or the end of the string
11  response_text: /(?s).+?(?=(?:HUMAN: |$))/
12      // The response text ends before "HUMAN:␣" or the end of the
          string
```

## B.2 Dataset Descriptions

**ShareGPT [1]**   The ShareGPT dataset contains multi-round conversations between users and GPT. We structure each conversation such that each user's query starts with 'HUMAN: ' and each GPT's response starts with 'GPT: '. The downstream task we conduct is to predict the user's intent and conversation topic based on user queries.

**ICLR [2]**   The ICLR dataset contains the reviews, author rebuttals, follow-up discussions, and final decisions of the papers submitted to ICLR 2024 [2]. Each review or reviewer's comment starts with

'Reviewer $n$' where $n$ represents the reviewer's identity, and each author rebuttal or discussion starts with 'Response'. The downstream task is to predict the research area of the paper based on the review and rebuttals.

**Water [52]**    The Water dataset contains reviews of water bottles. The columns are product_name, overall_rating, title, cleaned_review and the goal is to predict the current rating (column "rating") of the bottle, which takes values $1, 2, 3, 4, 5$.

**Arena [67]**    The Arena dataset contains pairs of human-model conversations. The columns are conversation_a, conversation_b and the goal is to predict which of the conversations are better (column "winner"), which takes values model_a, model_b, tie, "tie (bothbad)".

**Adult [6]**    The Adult dataset contains census data. The columns are age, workclass, fnlwgt, education, education-num, marital-status, occupation, relationship, race, sex, capital-gain, capital-loss, hours-per-week, native-country and the goal is to predict income (column "income"), which takes values $<= 50k$ or $> 50k$.

**Synthetic Datasets with Controllable Data Attributes**    We include two synthetic datasets[6] named **Synthetic Reviews** and the **Synthetic Grounding Dataset**. The reviews dataset has 4 fields, namely text, sentiment, emotion, and rating. The grounding dataset has 4 fields including two source documents, a query, and a response. We generate these datasets through a multi-step synthetic data generation process with GPT-4o wherein we verify whether the fields satisfy certain conditions, e.g., the reviews dataset is a 1:1 split of extreme negative and extreme positive reviews about products and the grounding dataset is a 1:1:1:1 split of relevant/irrelevant queries and consistent/inconsistent source documents. In particular, the reviews dataset is composed on only extreme reviews, either very positive or very negative. This differs from a typical review distribution and is unique to this particular dataset. Similarly, for the grounding dataset, we vary the samples along two axes, first on the consistency of the information between the sources and second, on the relevancy of the query to the sources. Each of the synthetic datasets is balanced in both their training and (downstream) test sets on these variations.

### B.3    Data Modeling of Each Dataset

Tables 3 and 4 shows the data modeling and structure rules of each dataset.

Table 3: Data modeling of each dataset

| Dataset | Sample $D$ | Sample Attributes | Node $O$ | Node Attributes |
|---|---|---|---|---|
| ShareGPT | a conversation | $a_1$: number of nodes
$a_2$: token length | a query/response | $v_1$: token length
$v_2$: topic
$v_3$: intent
$v_4$: speaker |
| ICLR | reviews & rebuttals
of a paper | $a_1$: number of nodes
$a_2$: token length
$a_3$: topic
$a_4$: final decision | a post from the
reviewer/author | $v_1$: token length
$v_2$: writer
$v_3$: review score |
| Water | water bottle
review | $a_1$: number of nodes
$a_2$: attitude | a column of
the tabular data | $v_1$: token length
$v_2$: review score |
| Arena | 2 conversations
to compare | $a_1$: number of nodes
$a_2$: winner | a column of
the tabular data | $v_1$: token length
$v_2$: winner |
| Adult | census information of an adult | $a_1$: age
$a_2$: workclass | a column of
the tabular data | $v_1$: token length
$v_2$: income |
| Reviews | annotated product review | $a_1$: number of nodes
$a_2$: rating | a review text | $v_1$: token length
$v_2$: rating |
| Grounding | 2 sources
and a QA pair | $a_1$: number of nodes
$a_2$: answer | a grounded response | $v_1$: token length
$v_2$: answer |

---

[6]https://www.kaggle.com/datasets/structpedataset/structpe-synthetic-datasets

Table 4: Structure rules of each dataset

| Dataset | Rules |
|---|---|
| ShareGPT | **[Alternate Speakers]** $\forall O_i, O_{i+1} : O_i[\texttt{Speaker}] \neq O_{i+1}[\texttt{Speaker}]$.
**[Format]** $O[\texttt{Speaker}] \in \{\texttt{User}, \texttt{AI Agent}\}$. If $O[\texttt{Speaker}] = \texttt{User}$, the text
starts with 'HUMAN: '; If $O[\texttt{Speaker}] = \texttt{AI Agent}$, the text starts with 'GPT: '. |
| ICLR | **[Format]** $O[\texttt{Writer}] \in \{\texttt{Author}, \texttt{Reviewer 1-9}, \texttt{Meta Reviewer}\}$.
If $O[\texttt{Writer}] = \texttt{Author}$, the text starts with 'Response:';
if $O[\texttt{Writer}] = \texttt{Reviewer n}$, the text starts with 'Reviewer n:' ($1 \leq n \leq 9$).
**[Format]** $O[\texttt{Review Score}] \in \{1, 3, 5, 6, 8, 10\}$.
**[Format]** $D[\texttt{Final Decision}] \in \{\texttt{Reject}, \texttt{Accept:poster}, \texttt{Accept:top5\%}, \texttt{Accept:top25\%}, \}$. |
| Water | **[Format]** $O[\texttt{Overall\_rating}] \in \{1.0, 1.1, 1.2, \ldots, 4.9, 5.0\}$.
**[Format]** $O[\texttt{Rating}] \in \{1, 2, 3, 4, 5\}$ |
| Arena | **[Format]** $O[\texttt{Winner}] \in \{\texttt{model\_a}, \texttt{model\_b}, \texttt{tie}, \texttt{tie (bothbad)}\}$.
O[Conversation_a] starts with "Question:" and has "Answer:" before somewhere in the following text.
O[Conversation_b] starts with "Question:" and has "Answer:" before somewhere in the following text. |
| Adult | **[Format]** $O[\texttt{income}] \in \{<= 50k, > 50k\}$.
**[Format]** Some of the columns are categorical (e.g. workclass, native-country).
**[Format]** Some of the columns are numerical (e.g. age, capital-gain). |
| Reviews | **[Format]** $O[\texttt{Rating}] \in \{1, 2, 3, 4, 5\}$ |
| Grounding | **[Format]** $O[\texttt{Consistency}] \in \{1, 2, 3, 4, 5\}$.
**[Format]** $O[\texttt{relevancy}] \in \{1, 2, 3, 4, 5\}$ |

### B.4 Evaluation Metrics of Each Dataset

The evaluation items of each dataset are summarized in Table 5.

For ShareGPT, in our experimental results, we show the semantic similarity of the node pair (query, response) as KND, show the distributional distance of the queries' token lengths as AM, and present the prediction accuracy of the conversation topics in downstream task performance.

For ICLR, we show the semantic similarity of the node pair (review, rebuttal) as KND, show the distributional distance of the reviews' token lengths as AM, and present the prediction accuracy of the paper's research area in downstream task performance.

## C  Additional Results on Struct-Bench

**Resource Costs**  All baselines are implemented and performed on a server with eight H100 GPUs. Running experiments took approximately 400 GPU hours.

**Implementation Details on Instruction Fine-tuning**  For both Instruct DP-FT and Instruct FT, we use the same instructions as those in the Random API of PE. We prepend the instructions to each training sample and fine-tune the foundation model for 20 epochs with batch size 32, weight decay 0.01, and learning rate $10^{-4}$. The fine-tuned model then generates new samples conditioned on the given instructions.

### C.1 Benchmarking DP Synthetic Data Generation Across Datasets

We present the results of benchmarking the DP synthetic data generation methods under different datasets with $\epsilon = 4$ in Table 6. We use GPT-2 for FT and DP-FT, and use GPT-4o for IF and PE.

### C.2 Benchmarking DP Synthetic Data Generation with Varying Privacy Budget

We illustrate the performance of PE and DP-FT on all metrics under different privacy budgets $\epsilon \in \{1, 2, 4, \infty\}$ on ShareGPT and ICLR datasets by radar plots in Figs. 11 and 12. Similar to §C.1, we use GPT-2 for FT and DP-FT, and use GPT-4o for IF and PE.

Table 5: Metrics for Different Datasets.

| Dataset | Structural Metrics | Non-structural Metrics | Downstream Task |
|---|---|---|---|
| ShareGPT | **CFG-PR**
**KND**
1. (query, response) pair
2. (response, query) pair
**AM**
1. number of nodes
2.query token length
3. response token length
4. topic
5. intent | 1. KNN-Precision
2. KNN-Recall | 1. topic prediction
2. intent prediction |
| ICLR | **CFG-PR**
**KND**
1. (review, rebuttal) pair
2. (rebuttal, comment) pair
3. (review, review) pair
from different reviewers
**AM**
1. number of nodes
2. review token length
3. rebuttal token length
4. Recommendation
5. final decision
6. topic | 1. KNN-Precision
2. KNN-Recall | topic prediction |
| Arena | **CFG-PR**
**KND**
1. (conversation_a, conversation_b) pair
**AM**
1. winner | 1. KNN-Precision
2. KNN-Recall | winner prediction |
| Water | **CFG-PR**
**KND**
1. (title, cleaned_review) pair
**AM**
1. attitude | 1. KNN-Precision
2. KNN-Recall | rating prediction |
| Adult | **CFG-PR**
**KND**
1. (native country, workclass) pair
**AM**
1. income | 1. KNN-Precision
2. KNN-Recall | income prediction |
| Reviews | **CFG-PR**
**KND**
1. (text, sentiment) pair
**AM**
1. review token length | 1. KNN-Precision
2. KNN-Recall | review label prediction |
| Grounding | **CFG-PR**
**KND**
1. (source1, source2) pair
**AM**
1. query relevancy | 1. KNN-Precision
2. KNN-Recall | query relevancy prediction |

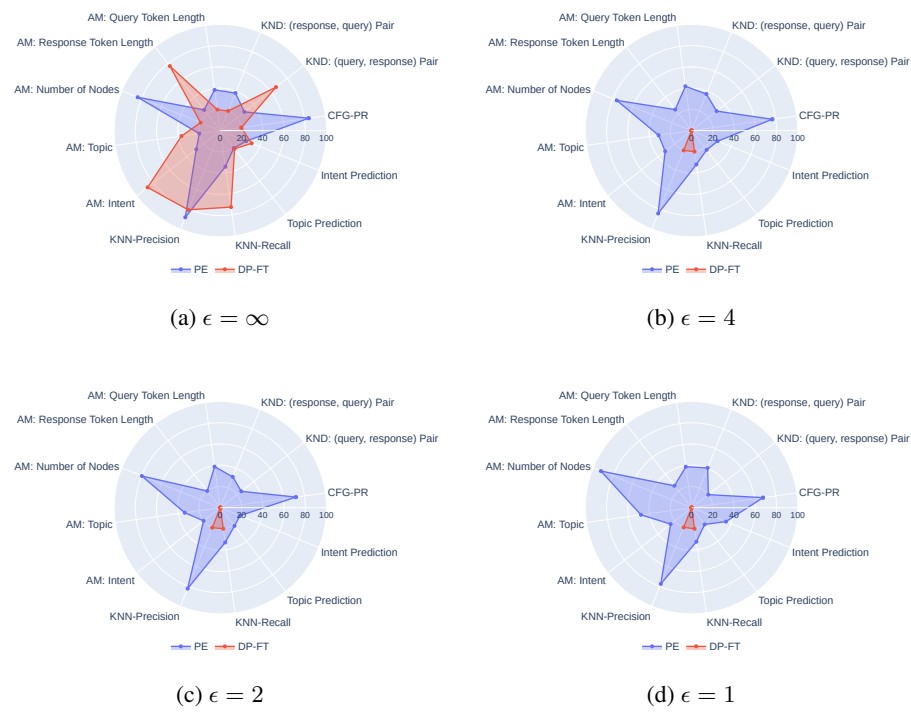

Figure 11: Performance of PE and DP-FT on all metrics under different privacy budgets on ShareGPT.

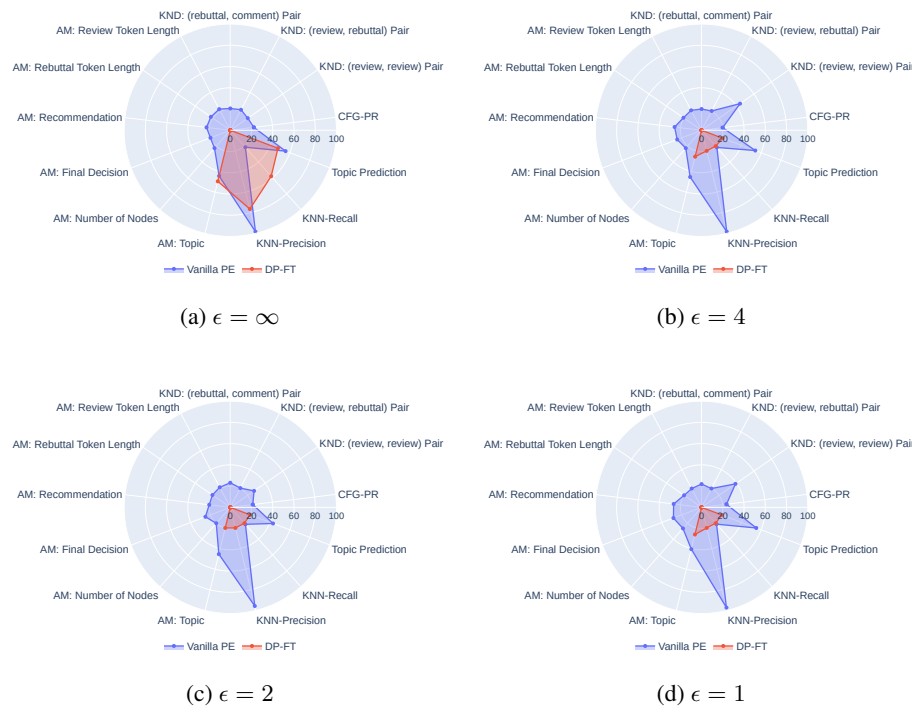

Figure 12: Performance of PE and DP-FT on all metrics under different privacy budgets on ICLR.

Table 6: DP synthetic data generation benchmarking results on Struct-Bench with $\epsilon = 4$

| Dataset | Baseline | Structural Metrics | | | Non-Structural Metrics | | DE |
| | | CFG-PR ↑ | KND ↓ | AM ↓ | KNN-Precision ↑ | KNN-Recall ↑ | Acc↑ |
|---|---|---|---|---|---|---|---|
| ShareGPT | IF ($\epsilon = 0$) | 0.8700 | 0.0635 | 43.8514 | 0.7217 | 0.2627 | 0.3754 |
| | FT ($\epsilon = \infty$) | 0.5378 | 0.0315 | 52.6984 | 0.7594 | 0.6588 | 0.3718 |
| | DP-FT | 0 | - | - | 0.0161 | 0.0000 | - |
| | PE | 0.8633 | 0.0660 | 38.1678 | 0.8050 | 0.1528 | 0.3816 |
| ICLR | IF ($\epsilon = 0$) | 0.1733 | 0.2582 | 204.7997 | 0.8400 | 0.0257 | 0.4715 |
| | FT ($\epsilon = \infty$) | 0 | - | - | 0.7056 | 0.4747 | 0.4584 |
| | DP-FT | 0 | - | - | 0.0000 | 0.0000 | 0.1806 |
| | PE | 0.1900 | 0.2599 | 240.9434 | 0.9800 | 0.0207 | 0.5218 |
| Water | IF ($\epsilon = 0$) | 1.0000 | 0.4222 | 0.1574 | 0.0000 | 0.0060 | 0.5485 |
| | FT ($\epsilon = \infty$) | 0 | - | - | 0.0000 | 0.0060 | - |
| | DP-FT | 0 | - | - | 0.0000 | 0.0060 | - |
| | PE | 1.0000 | 0.2877 | 0.0236 | 0.0000 | 0.0070 | 0.6130 |
| Arena | IF ($\epsilon = 0$) | 1.0000 | 0.1257 | 0.9395 | 0.0000 | 0.0090 | 0.3607 |
| | FT ($\epsilon = \infty$) | 0 | - | - | 0.0000 | 0.0060 | - |
| | DP-FT | 0 | - | - | 0.0000 | 0.0060 | - |
| | PE | 1.0000 | 0.1054 | 0.9193 | 0.0000 | 0.0070 | 0.3510 |
| Adult | IF ($\epsilon = 0$) | 1.0000 | 0.0290 | 0.0332 | 0.0030 | 0.0030 | 0.7920 |
| | FT ($\epsilon = \infty$) | 0 | - | - | 0.0030 | 0.0030 | - |
| | DP-FT | 0 | - | - | 0.0030 | 0.0030 | - |
| | PE | 1.0000 | 0.0042 | 0.0000 | 0.0030 | 0.0060 | 0.8017 |
| Reviews | IF ($\epsilon = 0$) | 1.0000 | 0.3510 | 0.4010 | 0.0334 | 0.0344 | 0.6000 |
| | FT ($\epsilon = \infty$) | 0 | - | - | 0.0020 | 0.0900 | 0.5400 |
| | DP-FT | 0 | 0.0020 | 0.0060 | 0.0020 | 0.0900 | 0.5600 |
| | PE | 1.0000 | 0.2495 | 0.0770 | 0.0290 | 0.0900 | 0.5400 |
| Grounding | IF ($\epsilon = 0$) | 1.0000 | 0.5800 | 0.6006 | 0.0500 | 0.0600 | 0.6400 |
| | FT ($\epsilon = \infty$) | 0 | - | - | 0.0290 | 0.0900 | 0.4000 |
| | DP-FT | 0 | - | - | 0.0430 | 0.0900 | 0.4000 |
| | PE | 1.0000 | 0.1435 | 0.4710 | 0.0300 | 0.0600 | 0.6000 |

## C.3 Benchmarking DP Synthetic Data Generation on ShareGPT using Llama2-7b

We illustrate the performance of PE, DP-FT, and Instruct DP-FT in Fig. 13 , where each dimension corresponds to a different metric from Struct-Bench. To better visualize differences in the performance of different methods, we scale the metrics in these radar plots as follows: We assign a score of 0 if CFG-PR=0 or a structure-related metric is not applicable for the dataset, and rescale the values of other metrics from 20 to 100, where 20 indicates the worst performance among all methods, and 100 indicates the performance upper bound the synthetic data can achieve (e.g., CFG-PR=1 or AM=0).

Fig. 13 shows that (1) DP-FT does not learn any structural information (CFG-PR); and (2) with instruction-guided conditional generation, Instruct DP-FT achieves similar performance to PE on most metrics and has a slight edge in terms of structure learning CFG-PR.

## D   Detailed analysis of the case study on PE

**Resource Costs**   All baselines are implemented and performed on a server with eight H100 GPUs. Running experiments took approximately 1000 GPU hours.

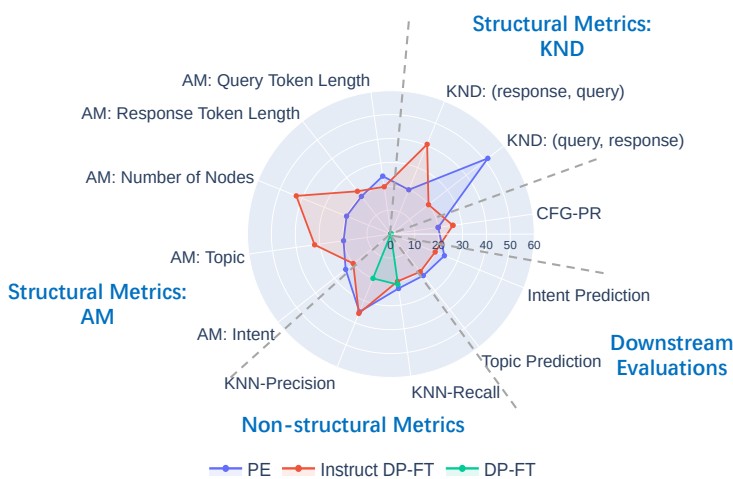

Figure 13: Performance of different baselines on ShareGPT using Llama2-7b with $\epsilon = 4$. With instruction-guided conditional generation, Instruct DP-FT achieves similar performance to PE on most metrics and has a slight edge in terms of CFG-PR.

### D.1 Analyzing Vanilla PE on ShareGPT Dataset

In this section, we analyze the performance of PE under the ShareGPT dataset according to our proposed benchmark. We further divide the metrics into semantic and statistic metrics, and the evaluation items for ShareGPT can be categorized in Table 7.

Table 7: Metrics for ShareGPT.

| | Statistic Metrics | Semantic Metrics | CFG-PR |
|---|---|---|---|
| **Structural Metrics** | AM: 1. number of statements 2. query token length 3. response token length | KND: 1. (query, response) pair 2. (response, query) pair AM: 1. topic 2. intent | CFG-PR |
| **Non-structural Metrics** | - | 1. KNN-Precision 2. KNN-Recall | - |
| **Downstream Tasks** | - | 1. topic prediction 2. intent prediction | - |

We illustrate and compare the performance of PE with privacy parameter $\epsilon \in \{1, 2, 4, \infty\}$ under structural semantic and statistic metrics in Figs. 14c and 14d respectively, and plot the CFG-PR and KNN-Precision & KNN-Recall in Figs. 14a and 14b. We do not include PE with $\epsilon = 0$ (that is, IF) as its CFG-PR is only 2% and thus its performance under structural metrics is unreliable.

As we can observe, only CFG-PR and KNN-Precision improve with the increase of $\epsilon$, while the value of KNN-Recall always keep around 0.35 and the performance under other semantic metrics and all statistic metrics does not necessarily increase with more relaxed privacy constraints. Additionally, CFG-PR drops below 60% when $\epsilon \leq 4$. Since downstream tasks also depend on structural information, we can conclude that PE mainly focuses on non-structural semantic quality of the synthetic samples, while suffers from poor performance on semantic diversity and structure-based properties.

### D.2 CFG Reformat Prompt

```
1   You are required to REFORMAT the provided conversation between a user
        and an AI agent in ChatGPT.  The format should be:
2     -User prompt must start with "HUMAN:␣", and ChatGPT response must
          start with "GPT:␣".
```

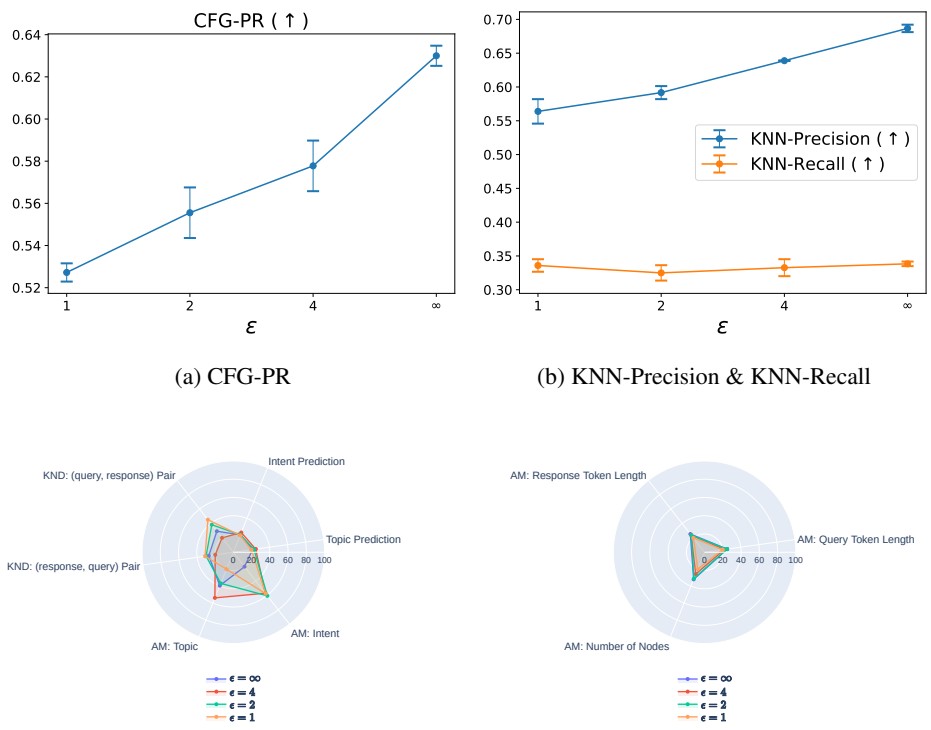

(a) CFG-PR

(b) KNN-Precision & KNN-Recall

(c) Performance on structural semantic metrics

(d) Performance on statistic metrics

Figure 14: Performance of Vanilla PE with different privacy guarantees under ShareGPT dataset

```
3       -The conversation may contain one or multiple rounds. Each round
            includes ONE user prompt and ONE ChatGPT response.
4       -User prompts and ChatGPT responses appear alternately.
5       -The conversation begins with a user prompts.
6   The reformatted conversation follows the following context-free
        grammar:
7       sharegpt: round (round)*
8       round: request response
9       request: "HUMAN:␣" user_string
10      response: "GPT:␣" gpt_string
11      user_string: /(?s).+?(?=(?:GPT: |HUMAN: |$))/
12      gpt_string: /(?s).+?(?=(?:GPT: |HUMAN: |$))/
13      %import common.WS
14      %ignore WS
15  Do NOT change the content of the conversation.
16  For example: If the input conversation is: "How␣are␣you?␣I'm␣fine."
        You should reformat it as "HUMAN:␣How␣are␣you?␣GPT:␣I'm␣fine."
```

### D.3 Further Analysis on CFG Reformat as Self-debugging

We compare the performance of vanilla PE and PE with CFG reformat on all evaluation items in Fig. 15. Self-debugging after voting directly reformats voted samples, which are taken as output or utilized as seeds in the next PE iteration without further selection, resulting in higher CFG-PR while lower performance on semantic and statistic properties.

### D.4 Improving Node Dependency (KND): Fix Format Token in Variation API

Key node dependency (i.e., KND) is an important semantic metric that measures the similarity of the node pair dependencies between the private and synthetic datasets. To improve KND, we fix the format tokens during blank-filling in variation API. Since nodes are recognized and separated by

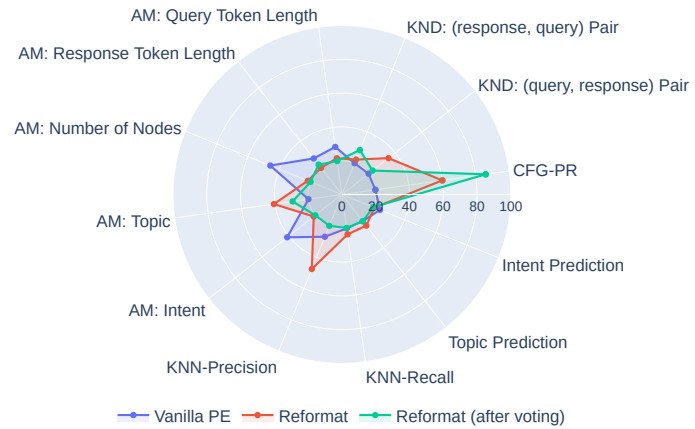

Figure 15: Performance of PE with CFG Reformat on ShareGPT with $\epsilon = 4$

format tokens in textual datasets, fixing the format tokens ensures that multiple nodes will not be mistakenly merged into one and thus helps to remain the original semantic meaning of each node, and therefore the node semantic dependencies. We compare the performance of vanilla PE and PE with fixed format token on KND on (query, response) and (response, query) pairs and CFG-PR in Fig. 16, where we consider two variants of our method: fix all the format tokens (shown as Fixed Token) and randomly fix 65% of the format tokens (shown as Fixed Selected Token). We can observe that our methods achieve better semantic performance on KND compared to vanilla PE, and Fixed Token outperforms since it keeps more node structures than Fixed Selected Token. Additionally, as fixing format tokens avoids node merging, it also improves the structural validity, i.e., CFG-PR.

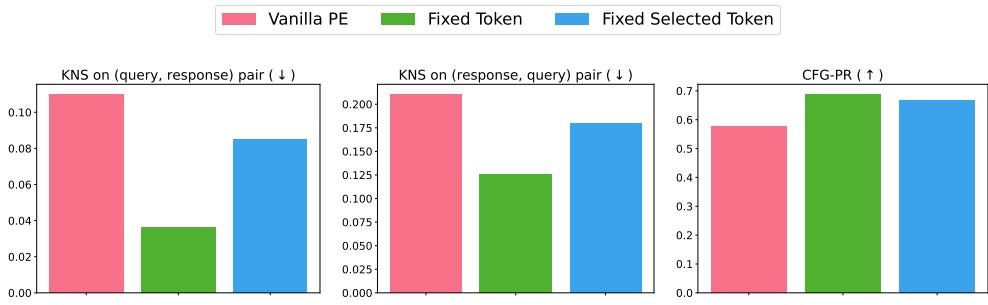

Figure 16: Performance of Vanilla PE and PE with fix token on CFG-PR and KND

We then compare the performance of vanilla PE and our methods on all metrics in Fig. 17. Since Fixed Token fixes all format tokens, the blank-filling process becomes less flexible, e.g., the number of nodes after blank-filling will never decrease, which is ensured by existing format tokens. Therefore, its performance in most statistic properties is worse than that of vanilla PE and Fixed Selected Token.

## D.5 Further Analysis on Node extraction & Auto-generation

To further examine the semantic diversity of the dataset, we adopt another metric Type to Token Ratio (TTR) [41] to provide auxiliary information. TTR measures diversity in the tokens used in the dataset by dividing the number of unique tokens by the total number of tokens in the dataset. A higher TTR suggests a more diverse vocabulary. Fig. 18 shows that Extract Query has a higher TTR than vanilla PE.

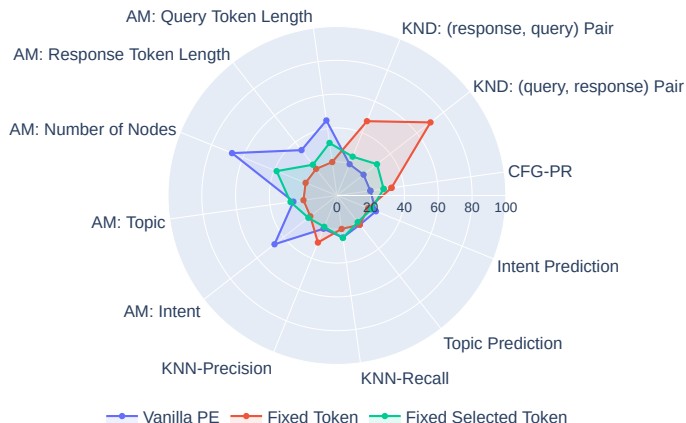

Figure 17: Performance of PE with Fix Format Token on ShareGPT with $\epsilon = 4$

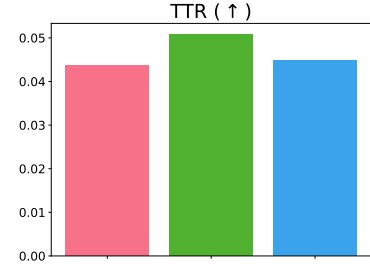

Figure 18: Performance of vanilla PE and PE with node extraction on Type to Token Ratio (TTR).

To illustrate the semantic quality and diversity of the synthetic dataset, we then focus on the embeddings of the generated sample, and draw them in a 2-dimensional plot after principal component analysis (PCA). As shown in Figs. 19a and 19b, the embeddings of vanilla PE and PE with query node extraction (blue dots) are drawn together with the embeddings of private data (yellow dots). We can easily observe that the embeddings of PE with query node extraction have more overlaps with the private data embeddings, indicating a higher sample semantic quality and diversity.

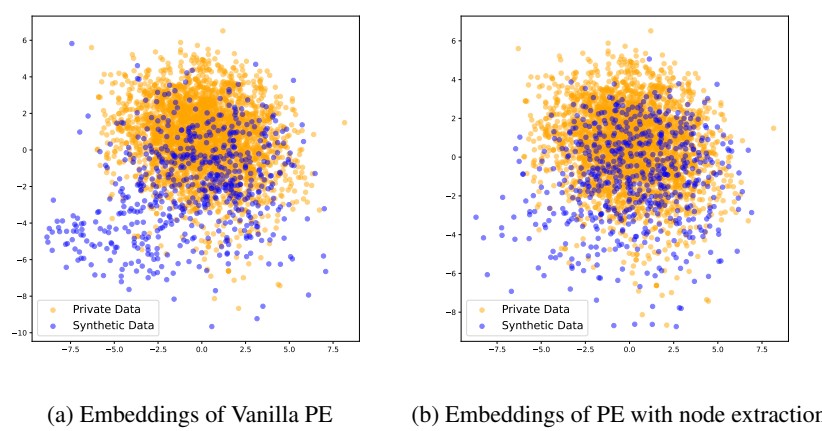

(a) Embeddings of Vanilla PE      (b) Embeddings of PE with node extraction

Figure 19: Embedding distributions of Vanilla PE and PE with node extraction.

We them compare the performance of vanilla PE and PE with node extraction on all metrics in Fig. 20, where we consider several variants of our method: extract all query nodes and auto-generate all response nodes (shown as Extract Query); combination of query node extraction, reformat before voting, and fix format token (Extract Query & Reformat & Fixed Token); combination of query node extraction, reformat before voting, and fix 65% format token (Extract Query & Reformat & Fixed Selected Token); combination of response node extraction, reformat before voting, and fix 65% format token (Extract Response & Reformat & Fixed Selected Token). We can observe that (1) Extract Query outperforms vanilla PE across most statistic properties, CFG-PR, and semantic properties including KNN-Precision, KNN-Recall, and KND on (response, query) pair. (2) Extract Query & Reformat & Fixed Selected Token outperforms or achieves similar performance to other node extraction variants on CFG-PR, most statistic and semantic metrics. This indicates that the combination of reformat and fix selected format tokens to node extraction improves CFG-PR and structural semantic performance without degrading statistic performance. (3) Extracting query nodes outperforms extracting response nodes, indicating that the type of nodes extracted significantly influences the synthetic data performance.

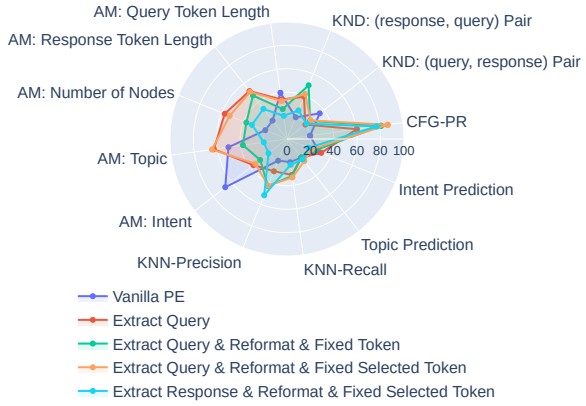

Figure 20: Performance of PE with Node Extraction on ShareGPT with $\epsilon = 4$

## D.6 Performance Comparison between Different Methods

We compare the performance of our proposed methods and some combinations of them according to our benchmark. Specifically, in Fig. 21, we illustrate the performance of vanilla PE; PE with CFG reformat; PE with fixed format token; combination of CFG reformat and fix format token (Fixed Token & Reformat); and combination of CFG reformat, fix partial format token, and query node extraction (Extract Query & Reformat & Fixed Selected Token). As we can observe, Extract Query & Reformat & Fixed Selected Token outperforms on structural validity CFG-PR, semantic properties KNN-Precision and KNN-Recall, and statistic properties AM on conversation round and response token length; while Fixed Token & Reformat outperforms mainly on semantic properties KND on (query, response) and (response, query) pair. As different methods focus on different aspects of the synthetic data, users can choose the method according to their practical needs. The algorithm design and analysis based on our benchmark also pave the way to propose a method that outperforms on all evaluation metrics, which we leave as a future work.

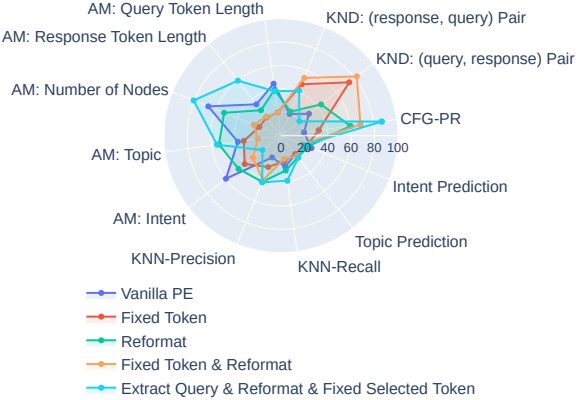

Figure 21: Performance of Different Methods on ShareGPT with $\epsilon = 4$

