# OpenReview forum: "Struct-Bench: A Benchmark for Differentially Private Structured Text Generation"
_NeurIPS.cc/2025/Datasets_and_Benchmarks_Track — NeurIPS 2025 Datasets and Benchmarks Track poster_

### Official Review · Reviewer_KLph · 2025-06-23

**Rating:** 5
**Confidence:** 3

**Summary:**

This paper introduces an evaluation protocol and benchmark for synthetic data that is i) structured and ii) generated with a DP mechanism. The focus is on natural language, but numerical/categorical features are supported too. The paper additionally contains a case study that shows how the benchmark uncovers shortcomings of existing algorithms for unstructured data and how those shortcomings can be fixed. Crucially, each sample's structure is described by a context-free grammar (CFG); this CFG is considered public information.

**Evaluation protocol:** The evaluation acts directly on datasets. Inputs are a real (private) dataset, a synthetic dataset, a schema describing each sample's structure in the form of a CFG, an optional set of features/node types whose correlations should be considered in metrics ("key nodes"), and a concrete instantiation of the metrics.

**Metrics:** All metrics focus solely on utility. The evaluation entails structural metrics, non-structural metrics, and downstream accuracy. The *structural metrics* measure i) the fraction of synthetic samples that follow the CFG schema, ii) how well correlations between key nodes in the synthetic data match correlations in the private data, and iii) how the distribution of certain attributes (e.g., number of tokens, number of nodes, or labels) in the synthetic data matches the distribution in the private data. *Non-structural* metrics only consider raw text; they use embedding similarity to find every sample's nearest neighbors in both the private and synthetic dataset. The actual metrics are then KNN-precision (how many synthetic samples are closer to a private sample than to a sample in their own dataset) and KNN-recall (how many private samples are closer to a synthetic sample than to a sample in their own dataset). Lastly, the *downstream evaluation* generates labels for all samples, fine-tunes an LLM on the synthetic data and labels, and evaluates the accuracy on a held-out test set of private data.

**Benchmark**: The benchmark uses 2 real-world structured datasets (ShareGPT and ICLR 2024 reviews), 3 real-world tabular datasets (Adult, water bottle reviews, Chatbot-Arena), and 2 synthetic datasets (product reviews and grounded Q&A). The authors evaluate 4 methods: private evolution, instruction following (generate text using a foundation model without private data access, only specifying the schema), DP fine-tuning a model on private data (without any structural constraints), and directly fine-tuning a model on the private data (without any privacy-preserving mechanism or structural constraints). Methods without structural constraints fail to generate data according to the CFG in almost all settings. No method performs well on all metrics, and there is no clear best one.

**Case study**: The case study primarily investigates the subpar performance of private evolution on a small Llama 2 model. The authors find that formatting synthetic data into the desired schema using an LLM (as a post-processing step) improves how well synthetic data matches the schema. They further propose a setting-specific modification to private evolution that improves semantic diversity (in terms of KNN-precision and KNN-recall).

**Additional Feedback:**

1. It could be useful to have a CFG user guide or some examples in the repo README, as the formalism might not be well-known for some users.
2. Why do structural metrics only consider pairs of key nodes, not higher-order relationships?
3. The reformatting example in L279--283 seems to not just reformat generated text, but also add new information. This might skew results, because the reformatting model is now technically a part of the method. Are there some safeguards against this in place? Are there other ways to enforce the CFG that avoid this issue?
4. The case study introduces a new metric on L324; why is this metric not used globally?
5. The paragraph on L39-48 is a bit dense; it could benefit from some illustrations or being more high-level.
6. L90 "Samples and objects can have attributes" should "objects" be "nodes"? In general, it would be clearer if terminology of nodes vs. objects was consistent.
7. Table 2 and Figure 3 mention $\epsilon = 4$; however, they also contain non-private fine-tuning (FT), which has $\epsilon = \infty$.
8. Why do the FT and DP-FT baseline not have Downstream Evaluation metrics for some datasets (ShareGPT, Water, Arena, Adult) in Tables 2/4? There are DE accuracies for some other datasets with 0% CFG pass rate (e.g., ICLR).

**Dataset Code Accessibility:**

Partly

**Dataset Code Comments:**

**Dataset**
- There is no dataset card or metadata on Kaggle yet. Adding a short description and the necessary metadata could help users to get started and make the benchmark more easily discoverable.
- In particular, the synthetic data is provided in a structured format (JSON files with different JSON attributes for every actual part of a sample). This makes sense, but adding a short example on how to generate actual natural-language training data from the structured data would aid usability.
- There exists a croissant file for the dataset, but it is pretty barebones (e.g., only linking to the zip archive); it particularly does not contain references to the actual json files containing the dataset.

**Code**
- The code contains documentation for most of the "getting started" parts.
- Some documentation is "hidden" inside the `structpe/dataset/ReadMe.md` file. This was not immediately clear to me; I would either merge that part with the global README or at least make a link.
- The code is overall documented sufficiently well to allow a new user to add their dataset synthetization method to the benchmark. However, this documentation is currently distributed across multiple files, and I had search for some run scripts to fully understand what to do. It would be easier for users if the global README contained a step-by-step guide on how to add a new dataset(/method), run the downstream evaluation, and obtain the final results.

**Ethical Considerations:**

No, there are no or only very minor ethics concerns

**Final Justification:**

The rebuttal resolved my two main concerns, the insufficient baselines and unclear description, and it also clarified and fixed most other minor issues. I think this is now a decent benchmark with good baselines, and hence raised my score.

**Limitations Weaknesses:**

**Insufficient baselines**: The benchmark effectively only has two baselines instead of four, because DP fine-tuning (DP-FT) and directly fine-tuning on the private data (FT) effectively do not work and provide no signal, as seen in Table 4. DP-FT never (0% of all cases) generates text that follows the schema; hence, there are no structural metrics for DP-FT at all. Directly fine-tuning on private data similarly never generates compliant outputs except on the ShareGPT dataset. This is particularly problematic for FT; a very strong non-private baseline is essential to give a rough upper bound on utility, and a DP fine-tuning baseline with non-trivial performance would also be a useful comparison point. A more reasonable baseline would be to enforce the schema when generating data with FT and DP-FT (either by providing the schema in the prompt, or by enforcing the schema during sampling). Notably, Section 4.1 even shows that simply providing the schema in the instructions works well for DP-FT. Therefore, I believe FT and DP-FT baselines that enforce the schema would provide a more fair and meaningful baseline.

In addition, as describe on L206-208, the benchmark uses different base models for different methods. This can be unfair, since schema-following likely correlates with model capabilities. It would be better to use GPT-2 for all methods, and then *additionally* GPT-4o for all methods that support it. Notably, GPT-4o can also be fine-tuned through the OpenAI API, so that model would also work for the FT baseline. Similarly, the case study uses yet another model (Llama2-7B), which is not good at following instructions/not very capable. I think the case study should be done on the actual benchmark setting, and hence also use the same model(s). The benchmark could further also include a PE variation that incorporates the improvements of Section 4.2 to have a stronger baseline.

**Lacking clarity of data format and metrics description**: Given only the paper, I found it hard to understand in which format datasets are actually expected to be. For example, L31-33 make it sound like the paper considers a mixture of categorical/numerical and free-text data, and the paragraph on L95 seems to imply that the CFG describes the overall structural relations between fields (not in the actual natural language content). Yet, my ultimate understanding is that the full evaluation pipeline only acts on strings, and that fully structured data (e.g., Adult) has to be converted into text (e.g., as JSON strings). This lack of clarity makes it hard for end-users to use the benchmark. I think it would be highly beneficial if i) the paper could make it clear in which format data should be provided, and ii) prescribed a unified way to convert structured data into free-text (e.g., via JSON formatting).

Moreover, the description of metrics (Section 2.1), especially for KND, AM, and downstream evaluation, is a bit vague/hard to understand. For some, the text does not define actual metrics but roughly references examples. In contrast, the downstream evaluation both introduces metrics and technical implementation details. It would be clearer if the section first introduced the metrics concisely, and then afterward describes how specific metrics are actually implemented. This would further benefit usability of the benchmark and made it harder to do "unfair" instantiations of metrics that benefit one method over another.

**Better justification of metrics needed**: In general, the metrics are overall sufficient, but feel particular and hand-crafted. It would be helpful if the paper featured some justification and limitations of the metrics. Furthermore, some metrics are fairly open-ended/underspecified. It would be helpful if the benchmark was more prescriptive in how metrics should be instantiatet. This would make the benchmark more useful, and avoid cases where different instantiations of a metric change the rank order of methods. Lastly, it would be helpful to verify the metrics with some qualitative examples.

**Potential issues in downstream evaluation**: I think that the downstream evaluation provides the most useful metric, but there are potential challenges in its design. One point are labels: If the private data has labels, then the synthetic data should have them too; there is thus no need to generate new labels. If the private data does not have labels, then the choice of labels has a very strong impact on downstream performance. Hence, I think there should be a safeguard against "hacking" a methods's performance by selecting a labeling function that particularly benefits that method (but not others).

**Summary**: While the benchmark is overall well-designed and targeting a relevant problem, the current baselines are insufficient, and current metrics are too underspecified. I believe that the benchmark is currently too impractical due to those points, but would be useful if they are resolved.

**Strengths Contributions:**

**Relevant setting and reasonable formalization**: The authors consider a relevant and practical setting of "messy" data that is structured and mixes numerical/categorical values with natural-language features. Using a CFG to is a clever layer of abstraction that makes it possible to work with very general structures yet still enables meaningful metrics. Although the formalism is slightly involved, the authors explain it reasonably well (e.g., Figure 1b).

**Flexible evaluation protocol**: The evaluation protocol is well-designed, in that it is broad enough to cover many datasets, while also being easy to instantiate. I believe that this helps this benchmark to stay useful for a longer time than other, more narrow benchmarks.

**Sensible choice of datasets**: The authors select a good variety of initial datasets, spanning a diverse set of use cases. The addition of two controlled synthetic datasets is also useful, as it allows debugging certain method's shortcomings. Similarly, the authors instantiate metrics well. I also like that the benchmark is designed with new datasets and methods in mind.

**Additional points**: I like the use of multiple runs and error bars where feasible (Figures 4 and 6). Additionally, the case study shows that this benchmark is also useful to identify (and mitigate) a method's potential shortcomings on structured data.

---

> ### Author Rebuttal · Authors · 2025-07-30
>
> **FT and DP-FT should also be given schema**
>
>
> Good point, we have included Instruct FT and Instruct DP-FT as baselines, where the schema is provided during both fine-tuning and sample generation processes. The results are shown in Table 1 below. Instruct DP-FT significantly outperforms DP-FT and even surpasses PE in structure-following (CFG-PR). Instruct FT achieves a better performance than PE and Instruct DP-FT. We agree this is a more reasonable baseline.
>
>
> **Table 1. Updated benchmark results**
>
>
> | Dataset | Baseline | CFG-PR ↑ | KND ↓ | AM ↓ | KNN-Precision ↑ | KNN-Recall ↑ | DE Acc↑ |
> |:---|:---|:---:|:---:|:---:|:---:|:---:|:---:|
> | ShareGPT | IF (ϵ = 0) | 0.03 | 0.07 | 41.86 | 0.64 | 0.31 | 0.28 |
> | | FT (ϵ = ∞) | 0.53 | **0.03** | 52.70 | 0.76 | **0.66** | 0.37 |
> | | Instruct FT (ϵ = ∞) | 0.66 | **0.04** | **30.70** | 0.83 | 0.54 | 0.37 |
> | | DP-FT | 0 | - | - | 0.02 | 0 | - |
> | | Instruct DP-FT | 0.60 | 0.18 | 32.59 | 0.77 | 0.31 | 0.35 |
> | | PE | 0.57 | 0.12 | 34.51 | 0.78 | 0.33 | **0.38** |
> | | PE & Reformat & Node Extraction | **0.89** | 0.13 | 30.91 | **0.88** | 0.43 | 0.36 |
> | ICLR | IF (ϵ = 0) | 0.09 | **0.11** | **207.62** | 0.66 | 0.28 | 0.39 |
> | | FT (ϵ = ∞) | 0 | - | - | 0.71 | **0.47** | 0.46 |
> | | Instruct FT (ϵ = ∞) | 0.17 | 0.16 | 208.37 | **0.77** | 0.29 | **0.51** |
> | | DP-FT | 0 | - | - | 0 | 0 | 0.18 |
> | | Instruct DP-FT | 0.14 | 0.22 | 237.52 | 0.49 | 0.18 | 0.40 |
> | | PE | 0.10 | 0.20 | 248.73 | 0.49 | 0.20 | 0.41 |
> | | PE & Reformat & Node Extraction | **0.49** | 0.21 | 214.75 | 0.76 | 0.44 | 0.39 |
>
>
> **Different base models for different methods. Better to use GPT2 for all methods + GPT-4o for all methods that support it.**
>
>
> Good point, we have added baselines for PE and IF using GPT2 so that all baselines are using the same base model (results in Table 1).
>
> That being said, we think adopting different base models is not necessarily methodologically incorrect—since our benchmark is agnostic to the underlying DP synthetic data algorithms, each DP algorithm should use its best, computationally-feasible version. This is done in many ML benchmarks, where baselines must comply with the benchmark API to participate.
>
>
> **Case study uses another model, should use the same model(s) as the table**
>
>
> We provide results for the algorithms in our case studies, i.e., Instruct DP-FT and PE with reformatting and node extraction, using GPT2 in Table 1.
>
> We will additionally keep the results on Llama3-8b-Instruct in Section 4.2, as our improvements on PE require instruction-following, and Llama3-8b-Instruct may be more accessible and cost-effective than GPT-4o for other researchers.
>
>
> **Benchmark should include a PE variation that incorporates improvements of Section 4.2**
>
>
> Table 1 includes a PE baseline with reformatting and node extraction using GPT2. The new baseline has better performance than PE and Instruct DP-FT, as expected.
>
>
> **In what format are datasets actually expected to be?**
>
>
> Apologies, the input data should indeed be in string format. For structured data with categorical or numerical values, we convert it to a JSON object where the attribute of each column is the key, and the actual value/text is its value. For instance, for a tabular row
>
>
> | Review    | Rating |
> | :--------: | :-------: |
> | I like it  | 5   |
>
>
> we convert it to: { “Review”: “I like it”, “Rating”: “5”}.
>
>
> **Vague description of metrics**
>
>
> We’ll add the precise definitions of KND, AM, and downstream evaluation as follows:
>
>
> Key Node Dependency (KND): KND measures the distributional distance of node pair dependencies between the synthetic and original data. For a key node pair $(O_i, O_j)$, let $C_{i,j}$ be the cosine similarity between their embeddings, and let $\omega_{C_{i,j}}$ and $\omega^\prime_{C_{i,j}}$ be the distributions of these similarities in the original and synthetic data, respectively. Then, KND is defined as:
> $$\text{KND}(O_i, O_j) = Dis(\omega_{C_{i,j}}, \omega^\prime_{C_{i,j}}),$$ where $Dis$ is the Wasserstein-2 distance.
>
>
> Attribute Mismatch (AM): AM calculates the distributional distance of a given attribute between the synthetic and original data. For attribute $a$, let $\omega_a$ and $\omega^\prime_a$ denote its distributions in the original and synthetic data, respectively. Then, AM is defined as:
> $$\text{AM}(a) = Dis(\omega_a, \omega^\prime_a).$$
> For distributional distance $Dis$, we use Wasserstein-2 distance for numeric attributes and total variation distance for categorical attributes.
>
>
> Downstream Evaluation (DE): Our evaluation pipeline consists of three steps—(1) Label Generation: obtaining labels for both synthetic and real data; (2) Model Training: fine-tuning a language model on the synthetic data and its labels; and (3) Evaluation: measuring label prediction accuracy on a held-out test set from the real data.
>
> **Better justification of metrics needed**
>
> We evaluate the synthetic data along three axes: structure, content, and downstream performance. We have defined our metrics (**boldfaced**) to provide coverage over these three categories.
>
> * Structure: We do not model structures that span multiple samples, so this is evaluated at the sample level. Within a sample, structure can be defined at the level of individual nodes (**AM**), groups of nodes (**KND**)*, and the whole sample (**CFG-PR**).
>
> * Content: Without using domain knowledge, we can assess content only through similarity to the real data. We do so by using standard metrics for semantic quality (**KNN-precision**) and diversity (**KNN-recall**) that can be easily applied to general data types (including tabular).
>
> * Downstream performance: We have posed a general classification framework for the downstream evaluation metric (**DE**);  the dataset holder should specify what kind of downstream task makes sense for the dataset.
>
> *Limitation: For KND, we measure dependencies between node pairs, and do not consider higher-order relationships since it is more computationally demanding.
>
> **How should metrics be instantiated?**
>
> For KND, we allow the user to specify key nodes. If not specified, all nodes parsed by CFG are treated as key nodes by default. To instantiate key nodes, we recommend users ask the question *“Which nodes are central to our downstream tasks, and which nodes are semantically related to them?”*. For example, key node pairs could be a query and response in a conversation dataset, or a review and its rating in a product review dataset. We’ve specified the key nodes of our datasets in Table1.
>
>
> For AM, users can specify semantic or statistical attributes. A guiding question is: *“Which data properties matter for our downstream tasks?”* Common semantic attributes include topic, intent, and sentiment; statistical attributes include token length (overall or per node). Original categorical/numerical values are also often relevant. The selected attributes for our datasets are detailed in Appendix A.4.
>
> **Verify metrics with qualitative examples**
>
> To illustrate the need for structural metrics, consider the synthetic ShareGPT sample: 'How are you? I’m doing well.' While semantically reasonable (thus scoring high on precision), it violates the required format, which mandates 'HUMAN:' before queries and 'GPT:' before responses. This highlights the necessity of structure-aware evaluation.
>
> For KND, consider the original sample: 'HUMAN: How are you? GPT: I’m good. Thanks.' Two synthetic variants are: (1) 'HUMAN: How are you? GPT: I’m doing well.' and (2) 'HUMAN: How are you? GPT: The weather is good.' The response is related to the query in both the original and the first synthetic sample. KND, which measures the similarity of node pair dependencies, assigns a lower score to the first, indicating closer alignment with the original.
>
>
> **Need safeguards against "hacking" a methods's performance via labeling**
>
> Performance hacking is a big problem (including in other ML benchmarks [1]), and may be difficult to resolve. We are considering two (imperfect) safeguards:
>
> 1. We could record submitting email addresses (e.g. from industry or academic domains) for both data and method contributors. This would expose if a method only excels on self-contributed datasets, but can be hacked via collusion.
>
> 2. Our leaderboard could rank methods by their median performance across all datasets, making it difficult to game the system without contributing a majority of datasets.
>
> [1] Eriksson, Maria, et al. "Can we trust ai benchmarks? an interdisciplinary review of current issues in ai evaluation." 2025.
>
>
> **Data and Code Comments**
>
>
> Thanks for the detailed feedback on our code and docs. We internally revised the data metadata and descriptions, and added both a step-by-step guide and a CFG user guide to the global README. In accordance with the regulations, we will update the public documentation after the rebuttal.
>
>
>
> **Why do structural metrics only consider pairs of key nodes?**
>
>
> Evaluating higher-order relationships is more computationally demanding and the dependency functions may vary across different scenarios. We will clarify.
>
> **Reformatting adds new information, may skew results**
>
>
> Reformatting is not part of the benchmark, but part of an improved PE method we propose to enhance the structure-following capability. Hence, it is allowed to add new (public) information.
>
>
> **Case study introduces a new metric (TTR). Use it everywhere?**
>
>
> We used TTR just as an auxiliary semantic diversity measurement. We believe KNN-recall is sufficient, and will remove TTR.
>
>
> **Para on L39-48 is dense**
>
>
> We’ll update the CFG design and key node selection processes in figure 1 to illustrate the pipeline of our benchmark more clearly.
>
>
> **Why do FT and DP-FT not have DE metrics for some datasets?**
>
>
> ICLR can be labeled as long as the final decision is in the synthetic data, whereas for other datasets like ShareGPT, labeling (e.g., topic of user queries) depends on valid structure to identify query segments, which we will clarify.

---

> > ### Comment · Reviewer_KLph · 2025-08-04
> >
> > I thank the authors for their very detailed response and the broad set of additional results.
> >
> > The additional experimental results resolve all my experiment issues in my review, and I believe the current results are solid baselines/provide great insights. I also thank the authors for clarifying the points of confusion. If the mentioned descriptions and justifications appear in an updated version of the paper, then I think it will be much easier for people to use the benchmark.
> >
> > Minor points:
> > 1. "That being said, we think adopting different base models is not necessarily methodologically incorrect": I agree with the authors in general; however, I believe that a more controlled setup (as is done now) is more sensible in the context of this particular paper.
> > 2. Regarding qualitative examples: I could have been more clear; my suggestion here is to dump a set of samples with particularly good/bad scores in the appendix, s.t. readers can gain intuition on how numerical values of metrics relate to actual data. There is no need to provide such examples in this discussion.
> > 3. "How to label samples for downstream evaluation" is still a challenge. I think the authors' suggestions are mitigations for adversarial "benchmark hacking". However, in my opinion, it could still be useful to be more prescriptive about how labels should be instantiated.

---

> > > ### Author Response · Authors · 2025-08-04
> > >
> > > Thank you very much for your careful reading of our paper. Your feedback has been very helpful for improving our work, and we very much appreciate your time and feedback. We will incorporate the provided empirical results, metric descriptions, and justifications into the paper after the rebuttal period (we are unable to update the paper yet).
> > >
> > > Finally, apologies for bringing this up, but if you find the updated results and responses satisfactory, we would be very grateful if you would consider raising your score at your convenience. Thank you for considering it.
> > >
> > > Here are our responses to your follow-up comments:
> > >
> > > **"That being said, we think adopting different base models is not necessarily methodologically incorrect": I agree with the authors in general; however, I believe that a more controlled setup (as is done now) is more sensible in the context of this particular paper.**
> > >
> > > We agree and will include a detailed description of the experimental setup, along with the provided experimental results, in the revised paper.
> > >
> > > **Regarding qualitative examples: I could have been more clear; my suggestion here is to dump a set of samples with particularly good/bad scores in the appendix, s.t. readers can gain intuition on how numerical values of metrics relate to actual data. There is no need to provide such examples in this discussion.**
> > >
> > > Good point. We will include several real examples with their corresponding scores in the appendix in the updated version of the paper.
> > >
> > > **"How to label samples for downstream evaluation" is still a challenge. I think the authors' suggestions are mitigations for adversarial "benchmark hacking". However, in my opinion, it could still be useful to be more prescriptive about how labels should be instantiated.**
> > >
> > > Thank you for pointing this out. To instantiate labels for downstream evaluation, we recommend that users first determine whether the label should be extracted directly from the sample or generated. Extracted labels typically include original categorical or numerical values—such as income level in our Adult dataset or paper decision in the ICLR dataset. If the label is to be generated, we suggest considering the question: *“What is a data property that is present and can be inferred from each node in a sample and is relevant to our downstream tasks?”* Generated labels are usually semantic attributes of the samples, such as topic, intent, or sentiment. To discourage prompt gaming by data uploaders, we provide a suggested labeling prompt template: *“For each sample in _____ scenarios, label the _____ property. Possible answers include {____, _____, ____, …}.”* The specific labels used for our datasets are detailed in Appendix A.4.

---

> > > > ### Comment · Reviewer_KLph · 2025-08-05
> > > >
> > > > I thank the authors for their reply. I think a "recipe" for generating labels such as the one described makes sense, and I suggest adding that to the README/benchmark description.
> > > >
> > > > With that, I have no further questions and comments.

---

> > > > > ### Author Response · Authors · 2025-08-05
> > > > >
> > > > > Thank you for the suggestion. We will incorporate the label generation instructions into the benchmark description in the updated version of the paper.

---

### Official Review · Reviewer_snJX · 2025-07-03

**Rating:** 5
**Confidence:** 4

**Summary:**

This submission concerns a system to evaluate synthetic data which can be structured by a context-free grammar (CFG).  The code is open source and also allows to keep a leader board for easier comparison of results.

**Additional Feedback:**

Not applicable.

**Dataset Code Accessibility:**

Yes

**Dataset Code Comments:**

The shared github is well documented.

The shared github has on some pages the title "struct-PE" rather than "struct-Bench".

**Ethical Comments:**

As far as i can verify there are no ethical concerns.
The methodology seems adequate.
The paper doesnt claim security/privacy, it is the responsibility of the evaluated systems to be secure.
Likewise, other aspects such as fairness and human rights are much more the responsibility of the evaluated systems.  These are not elements evaluated by the proposed benchmark.

**Ethical Considerations:**

No, there are no or only very minor ethics concerns

**Final Justification:**

I thank the authors for their comments which are largely consistent with what i already understood.  I maintain my score.

**Limitations Weaknesses:**

The evaluation seems to be applicable to both differentially private and other synthetic data as it mainly assesses the quality of the synthetic data and for privacy mainly relies on privacy guarantees offered by the evaluated system.  I guess the approach also would work for other forms of statistical privacy.  If true, maybe it is useful to mention that.
While some parts of the code remain a bit unclear to me, compared to other submissions the code is well elaborated and meaningful documentation is available.

**Strengths Contributions:**

The contribution is relevant, interesting, novel and likely significant.
The section discussing the relation to existing work is rather short, but I don't know of important missing related work (I'm not an expert in the synthetic data evaluation field).  The text is well-written and organized.
While the leader board is less novel, it is a useful addition.  The use of CFG seems really interesting.

---

> ### Author Rebuttal · Authors · 2025-07-30
>
> We sincerely thank the reviewer for their thoughtful feedback. Here are our responses to your concerns and questions:
>
> **The evaluation seems to be applicable to both differentially private and other synthetic data as it mainly assesses the quality of the synthetic data and for privacy mainly relies on privacy guarantees offered by the evaluated system. I guess the approach also would work for other forms of statistical privacy. If true, maybe it is useful to mention that.**
>
> You are correct, the framework does apply to other forms of privacy-preserving synthetic data (e.g., under frameworks like probably approximately correct privacy, quantitative information flow [1], statistic maximal leakage [2], distribution privacy [3], etc.) We will add a discussion of this point.
>
> [1] Smith, Geoffrey. "On the foundations of quantitative information flow." International Conference on Foundations of Software Science and Computational Structures. Berlin, Heidelberg: Springer Berlin Heidelberg, 2009.
>
> [2] Wang, Shuaiqi, Zinan Lin, and Giulia Fanti. "Statistic Maximal Leakage." 2024 IEEE International Symposium on Information Theory (ISIT). IEEE, 2024.
>
> [3] Suri, Anshuman, and David Evans. "Formalizing and estimating distribution inference risks." arXiv preprint arXiv:2109.06024 (2021).
>
>
>
> **While some parts of the code remain a bit unclear to me, compared to other submissions the code is well elaborated and meaningful documentation is available. The shared github has on some pages the title "struct-PE" rather than "struct-Bench".**
>
> Thank you for pointing this out. We have prepared a more detailed step-by-step guide in the global README. In accordance with the regulations, we will update the public documentation after the rebuttal period.

---

> > ### Comment · Reviewer_snJX · 2025-08-06
> >
> > Thanks for your rebuttal.  I think we agree, i have no further questions.

---

### Official Review · Reviewer_UZTu · 2025-07-04

**Rating:** 5
**Confidence:** 3

**Summary:**

This paper introduces Struct-Bench, a comprehensive evaluation framework and benchmark designed to assess the quality of differentially private synthetic data for structured datasets that include natural language fields. The authors propose representing data structure using context-free grammars to evaluate both syntactic and semantic properties of the synthetic datasets. They release a set of benchmarks and evaluate several DP generation methods. Additionally they conduct a case study showing how Struct-Bench can guide improvements in existing generation methods, achieving better structural validity and semantic diversity

**Dataset Code Accessibility:**

Yes

**Dataset Code Comments:**

The dataset and accompanying benchmark code are publicly accessible at the provided project website.

**Ethical Considerations:**

No, there are no or only very minor ethics concerns

**Final Justification:**

The weaknesses I addressed have been properly addressed.

**Limitations Weaknesses:**

* Struct-Bench requires users to manually define CFGs and key nodes. This could weaken usability for new datasets or practitioners lacking domain expertise. It could be more helpful to create CFGs and key nodes more automatically without an effort from domain experts.
* CFGs are inherently limited in expressiveness. In particular, they struggle to represent long-range or cross-field dependencies that commonly arise in real-world datasets. For example, datasets with conditional constraints such as ``if the user is under 18, the field 'employment status' must be ‘unemployed’'' involve semantic or logical conditions that are beyond the modeling capacity of CFGs. These limitations raise concerns about whether Struct-Bench can generalize to datasets with rich, context-sensitive dependencies.

**Strengths Contributions:**

* The benchmark is timely and novel. It observes an unobserved factor of dataset--structure of a dataset with natural language. This was also not addressed by traditional metrics like FID, MMD or downstream model's performance.
* The paper points out that the existing DP synthetic generation algorithms cannot capture the structure of given real-world datasets, which could be a new challenge in this field.
* From observation on applying Struct-Bench, the paper demonstrates how Struct-Bench can guide improvements in PE, increasing structure validity and semantic coverage.

---

> ### Author Rebuttal · Authors · 2025-07-30
>
> We sincerely thank the reviewer for their thoughtful feedback. Here are our responses to your concerns and questions:
>
> **Struct-Bench requires users to manually define CFGs and key nodes. This could weaken usability for new datasets or practitioners lacking domain expertise. It could be more helpful to create CFGs and key nodes more automatically without an effort from domain experts.**
>
> This is a good point; although the CFG specification only needs to be done manually once–namely, when the dataset is onboarded–manual grammar specification can be labor-intensive and error-prone [1]. It may be possible to use LLMs to summarize the structure and specify the CFG. In principle, one could even use an LLM to propose nodes that are semantically correlated.  We did not explore this level of automation in the current paper, in part because automated grammar extraction is an active area of research, with very recent work on LLM-assisted grammar extraction  (e.g., [2]). That being said, we agree that automating this aspect of the benchmark would be a useful next step.
>
> [1] Le, Triet HM, Hao Chen, and Muhammad Ali Babar. "Deep learning for source code modeling and generation: Models, applications, and challenges." ACM Computing Surveys (CSUR) 53.3 (2020): 1-38.
>
> [2] Torkamani, Mohammad Jalili. "Kajal: Extracting Grammar of a Source Code Using Large Language Models." arXiv preprint arXiv:2412.08842 (2024).
>
> **CFGs are inherently limited in expressiveness. In particular, they struggle to represent long-range or cross-field dependencies that commonly arise in real-world datasets. For example, datasets with conditional constraints such as ``if the user is under 18, the field 'employment status' must be ‘unemployed’'' involve semantic or logical conditions that are beyond the modeling capacity of CFGs. These limitations raise concerns about whether Struct-Bench can generalize to datasets with rich, context-sensitive dependencies.**
>
> This is also a good observation, thanks for pointing it out.  We considered modeling datasets with a context-sensitive grammar (CSG), but ultimately decided to use a CFG to balance usability with informativeness. That is, specifying a CSG requires more domain knowledge and detail, thus making it a more onerous (and potentially error-prone) process than specifying a CFG. On the other hand, as you pointed out, CFGs are less expressive. Thus, instead of expressing semantic dependencies as a hard constraint (i.e., with a CSG), we empirically evaluate semantic dependencies using the key node dependency (KND) metric. KND is easy to implement and does not require as much domain knowledge as fully specifying a CSG, but it captures correlations in the data statistically. We will add a discussion of this point to the text.

---

> > ### Comment · Reviewer_UZTu · 2025-08-08
> >
> > I thank the authors for the rebuttal.
> > My concerns have been properly addressed. I have no further questions.

---

### Official Review · Reviewer_tyxQ · 2025-07-04

**Rating:** 5
**Confidence:** 3

**Summary:**

Struct-Bench introduces a novel evaluation framework for differentially private (DP) synthetic data derived from structured datasets with natural language fields (e.g., conversation logs, reviews). Traditional metrics fail to evaluate such hybrid data, so Struct-Bench uses a Context-Free Grammar (CFG) to capture structure and semantic dependencies. The benchmark includes diverse datasets, multiple DP generation baselines (e.g., PE, DP-FT), and evaluation metrics covering structure, semantics, and downstream utility. Experiments reveal significant weaknesses in current DP generation methods and demonstrate how Struct-Bench can guide improvements.

**Dataset Code Accessibility:**

Yes

**Ethical Considerations:**

No, there are no or only very minor ethics concerns

**Final Justification:**

The authors have well addressed my concerns. I will keep my positive score.

**Limitations Weaknesses:**

Primarily tailored to structured + text datasets; does not fully address unstructured or purely numerical data.

Current DP methods still struggle to capture semantic variety under strict privacy constraints (e.g., low ε).

**Strengths Contributions:**

First benchmark to focus on evaluating DP synthetic data for structured datasets containing natural language.

Combines structural, non-structural, and downstream task metrics (e.g., CFG-PR, KNN Recall, prediction accuracy).

Demonstrates concrete improvements to DP generation methods (e.g., PE) using Struct-Bench insights.

---

> ### Author Rebuttal · Authors · 2025-07-30
>
> We sincerely thank the reviewer for their thoughtful feedback. Here are our responses to your concerns and questions:
>
> **Primarily tailored to structured + text datasets; does not fully address unstructured or purely numerical data.**
>
> Our benchmark also supports unstructured or numerical data. All non-structural metrics & downstream evaluations can be applied to evaluate unstructured synthetic data, while all the metrics in our benchmark can be adopted to evaluate purely numerical data. In fact, to illustrate the generality of Struct-Bench, we included the Adult dataset in our evaluation, which contains purely numerical data.
>
> **Current DP methods still struggle to capture semantic variety under strict privacy constraints (e.g., low ε).**
>
> This is true; however, we would like to emphasize that our aim in this paper is to design a systematic method for *evaluating* current DP methods (i.e., our main goal is not DP algorithm design). Hence, the fact that current DP methods struggle to capture semantic variety is in some sense the point of our work—Struct-Bench surfaces these shortcomings, which can be useful for motivating future research.   Our case study  conducts a preliminary exploration of the solution space, but  the  key point of the paper is that Struct-Bench allows us to pinpoint the drawbacks of the current DP methods. We will clarify this in the main text.

---

> > ### Comment · Reviewer_tyxQ · 2025-08-05
> > **maintain my posiitive socre**
> >
> > Thank the authors for the clarifications, and I will maintain my posiitive socre.

---

### Comment · Area_Chair_1B1W · 2025-08-03
**Reminder for Reviewer-Author discussions**

Dear reviewers,

Thank you so much for all your time and effort supporting NeurIPS!

If you haven't yet, please take a moment to read through the author's rebuttal. The reviewer-author discussion period is crucial for ensuring a fair and comprehensive evaluation of their work. If the rebuttal addresses your concerns, please acknowledge this and adjust your scores accordingly. If not, please let them know which concerns remain and if you have any follow-up questions. Your thoughtful feedback will help authors improve their scholarship and propel our field forward.

I know this is a busy time, and really appreciate your effort.

Best Regards
Area Chair

---

### Decision · Program_Chairs · 2025-09-18

**Decision:**

Accept (poster)

**Comment:**

Struct-Bench introduces a novel evaluation framework for differentially private (DP) synthetic data derived from structured datasets with natural language fields.Traditional metrics fail to evaluate such hybrid data, so Struct-Bench uses a Context-Free Grammar (CFG) to capture structure and semantic dependencies. The benchmark includes diverse datasets, multiple DP generation baselines (e.g., PE, DP-FT), and evaluation metrics covering structure, semantics, and downstream utility. Experiments reveal significant weaknesses in current DP generation methods and demonstrate how Struct-Bench can guide improvements.

Strengths:

1.	First benchmark to focus on evaluating DP synthetic data for structured datasets containing natural language.
2.	The paper points out that the existing DP synthetic generation algorithms cannot capture the structure of given real-world datasets, which could be a new challenge in this field.
3.	This paper demonstrates how Struct-Bench can guide improvements in PE, increasing structure validity and semantic coverage.

Weaknesses:

1.	Struct-Bench requires users to manually define CFGs and key nodes. This could weaken usability for new datasets or practitioners lacking domain expertise.
2.	CFGs are inherently limited in expressiveness. In particular, they struggle to represent long-range or cross-field dependencies that commonly arise in real-world datasets. For example, datasets with conditional constraints such as ``if the user is under 18, the field 'employment status' must be ‘unemployed’'' involve semantic or logical conditions that are beyond the modeling capacity of CFGs. These limitations raise concerns about whether Struct-Bench can generalize to datasets with rich, context-sensitive dependencies.
3.	Insufficient baselines.
4.	Lacking clarity of data format and metrics description. Better justification of metrics needed, it would be helpful if the paper featured some justification and limitations of the metrics.

Reasons:

The benchmark is timely and novel.

Discussions:

After the Reviewer KLph pointed the Insufficient baselines, authors add more baselines, which has been recognized by the reviewer.
Reviewer KLph also raised the questions and lack clarity about the metrics descriptions. Authors answers and improves about that.